# Enhancer-driven 3D chromatin domain folding modulates transcription in human mammary tumor cells

Silvia Kocanova[1], Flavien Raynal[1], Isabelle Goiffon[1], Betul Akgol Oksuz[3], Davide Baú[5], Alain Kamgoué[1], Sylvain Cantaloube[1], Ye Zhan[3], Bryan Lajoie[3], Marc A Marti-Renom[5,6,7,8], Job Dekker[3,4], Kerstin Bystricky[1,2]

The genome is organized in functional compartments and structural domains at the sub-megabase scale. How within these domains interactions between numerous cis-acting enhancers and promoters regulate transcription remains an open question. Here, we determined chromatin folding and composition over several hundred kb around estrogen-responsive genes in human breast cancer cell lines after hormone stimulation. Modeling of 5C data at 1.8 kb resolution was combined with quantitative 3D analysis of multicolor FISH measurements at 100 nm resolution and integrated with ChIP-seq data on transcription factor binding and histone modifications. We found that rapid estradiol induction of the progesterone gene expression occurs in the context of preexisting, cell type-specific chromosomal architectures encompassing the 90 kb progesterone gene coding region and an enhancer-spiked 5′ 300 kb upstream genomic region. In response to estradiol, interactions between estrogen receptor α (ERα) bound regulatory elements are reinforced. Whereas initial enhancer–gene contacts coincide with RNA Pol 2 binding and transcription initiation, sustained hormone stimulation promotes ERα accumulation creating a regulatory hub stimulating transcript synthesis. In addition to implications for estrogen receptor signaling, we uncover that preestablished chromatin architectures efficiently regulate gene expression upon stimulation without the need for de novo extensive rewiring of long-range chromatin interactions.

## Introduction

Rapid cellular responses to external stimuli rely on regulation of gene expression. Among numerous steps required for this response, the spatial organization of the genome is known to modulate DNA accessibility to the transcriptional machinery and to promote contacts between genes and distant regulatory DNA elements such as enhancers (Kooren et al, 2007; Gheldof et al, 2010;

Smith et al, 2016; Paliou et al, 2019; Oudelaar et al, 2021). In the past two decades, different levels of 3D folding of the genome have been described thanks to ever improving technologies from population-based contact frequencies to single-cell imaging at high resolution (Gibcus & Dekker, 2013; Kim & Shendure, 2019; van Steensel & Furlong, 2019). Entire chromosomes adopt differential conformations as a function of transcriptional competence, for example, between the active and inactive X-chromosomes (Boninsegna et al, 2022). All chromosomes are organized into active and inactive compartments composed of mega-base domains, and, at the scale of 10–100 s of kb, can form topologically associating domains (TADs). TADs reflect areas of increased contact probabilities between DNA elements, but their intrinsic organization is highly complex, variable between cells in the population, and their relevance subject to debate (Sikorska & Sexton, 2020). There is no doubt, however, that numerous genes and their regulatory elements are located within a single TAD, and that TAD boundaries reduce the probability of long-range contact between elements and genes located on different sides. How this relates to controlled transcriptional activity remains to be fully understood.

3D chromatin folding reorganizes during differentiation and development reflecting changes in transcriptional activity (Tan et al, 2021). Estradiol (E2) signaling is a well-studied paradigm for transcriptional regulation in mammalian cells. This hormone exerts essential pleiotropic actions during development and differentiation and is best known for its role in the function of reproductive tissues of both males and females. A major role of estrogens is to modulate the transcriptional status of target genes, with these actions being transduced through a specific nuclear receptor, the estrogen receptor α (ERα). ERα is a major determinant of tumor growth in about 80% of breast cancers in which it controls cell cycle genes such as cyclin D1 (CCND1) and differentiation genes such as the progesterone receptor gene (*PGR*) and the ERα coding gene itself (ESR1). Estrogens and hormones, in general, control transcription of hundreds of genes in the eukaryotic nucleus. It was shown that changes in gene expression are accompanied by modifications in global genome structure in ER-expressing cells

[1]Molecular, Cellular and Developmental Biology Unit (MCD), Centre de Biologie Integrative (CBI), University of Toulouse, UPS, CNRS, Toulouse, France [2]Institut Universitaire de France (IUF), Paris, France [3]Department of Systems Biology, University of Massachusetts Chan Medical School, Worcester, MA, USA [4]Howard Hughes Medical Institute, Chevy Chase, MD, USA [5]Centre Nacional d'Anàlisi Genòmica (CNAG), Barcelona, Spain [6]Genome Biology Program, Centre de Regulació Genòmica (CRG), Barcelona, Spain [7]Universitat Pompeu Fabra (UPF), Barcelona, Spain [8]Institució Catalana de Recerca i Estudis Avançats (ICREA), Barcelona, Spain

Correspondence: kerstin.bystricky@univ-tlse3.fr; Job.Dekker@umassmed.edu; martirenom@cnag.eu

(Le Dily et al, 2014; Zhang et al, 2020). Yet, individual TADs were maintained during hormone-induced activation. Le Dily et al (2014) proposed that the formation of structural regulons could promote coordinated expression of several genes. Not all of these genes would directly be estrogen regulated because ERα target genes do not need to colocalize to be activated (Kocanova et al, 2010a). Numerous estrogen receptor binding sites (ERBSs) are not only present at some promoters of ERα target genes but are also found located at 10–100 kb distance from target genes. A role in attracting co-factors to connect these distant elements and target genes has been proposed for a subset of ER-regulated genes (Kininis et al, 2007) and Chia-PET analysis suggests increased contacts (Fullwood et al, 2009). We thus asked if TADs are gene specific, reflecting and/ or contributing to regulation of the gene and other regulatory elements within a domain.

Here, we integrate data from high resolution 5C, 3D FISH, ChIP-seq, and computational modeling to analyze structural features of genomic domains containing several ERα target genes in two human breast cancer cell lines. Integration of data from such orthogonal experimental approaches enable establishing models of nuclear organization and defining statistically relevant structure/ function relationships (Nir et al, 2018; Abbas et al, 2019; Szabo et al, 2020; Boninsegna et al, 2022). We show that folding of ERα target gene domains differs between the silent and the activatable form of the ESR1 and PGR genes. In ERα-positive MCF7 cells, the pre-established domain conformation reorganizes upon E2-induced stimulation bringing distant ERα-bound enhancer elements in proximity of the gene body.

# Results

### High-resolution 3D maps of genomic domains encompassing estrogen-sensitive genes are cell type-specific

To determine whether the 3D organization of chromatin domains correlates with transcriptional status, we generated 5C chromatin interaction maps of ERα target genes whose transcriptional status differs in two human breast cancer cell lines with distinct tumor origins, MCF7 and MDA-MB-231 (Fig 1A). These cell lines are representative of breast cancer (BC) types: MCF7 cells express the ERα+ and their growth is hormone dependent. MDA-MB-231 cells are triple negative, hence do neither express the ERα, the PGR, nor the human epidermal growth factor receptor 2 (Her2) and their growth is independent of hormones. We selected 0.6–1.3-Mb domains around four ER-regulated genes: the estrogen receptor gene (ESR1) located on chromosome 6, the growth regulation by estrogen in breast cancer 1 gene (GREB1) located on chromosome 2, the cyclin D1 gene (CCND1) and the PGR on chromosome 11, and determined their 3D conformation using 5C (Dostie et al, 2006) in MCF7 and MDA-MB-231 cell lines (Fig 1B). Genes were chosen based on the transcriptional status and estrogen responsiveness (Giamarchi et al, 1999; Honkela et al, 2015). Expression levels of the analyzed genes were confirmed by RT–qPCR (Fig S1A). PGR and ESR1 were inducible in MCF7 cells and silent in MDA-MB-231 cells. GREB1 and CCND1 were constitutively

transcribed in MDA-MB-231 cells, and were hormone-inducible in MCF7 cells (Fig S1A).

We used Chromosome Conformation Capture Carbon Copy (5C) with an alternating primer design (Dostie et al, 2006; Kim & Dekker, 2018) to assess contact frequencies at a resolution of 1.8 kb from cells growing in a hormone-stripped medium for 3 d (Fig 1A). 5C contact frequency maps revealed a high degree of similarity of the overall organization of the studied domains between the two phenotypically distinct cell lines. Within the CCND1 and GREB1 gene domains, only a few architectural features were cell type-specific in agreement with the fact that these genes are expressed in both cell lines (Figs 1B and S1). In contrast, significant conformational differences were detected for the PGR and ESR1 gene domains. PGR and ESR1 were silent in MDA-MB-231 cells, but were transcriptionally active in MCF7 cells (Fig S1). In MCF7 cells, the PGR domain features two TADs with a clear boundary at the 3'end of the TRPC6 gene. TRPC6 codes for a transient receptor channel complex overexpressed in BC cells as compared with non-tumorous cell lines (Jardin et al, 2018) (Fig S1B). This TAD organization was not present in MDA-MB-231 cells in which numerous weak long contact frequencies characterize the entire domain and no TAD boundary was detected (Fig S1B). The PGR gene located within a 450-kb region which corresponds to the first TAD (TAD1), revealed slightly stronger interaction frequencies at long distances in MCF7 compared with MDA-MB-231 cells (Figs 1C and S1B). The ESR1 gene domain featured greater long-distance interactions in MDA-MB-231 cells compared with MCF7 cells (Figs 1C and S1C green arrowhead), interactions which appeared to stem from loci flanking the gene itself and the 5' end of SYNE1 (Fig 1B). A domain boundary is present within the SYNE1 gene. SYNE1 is rarely transcribed in MCF7 cells (proteinatlas.org). In MDA-MB-231 cells, SYNE1 is expressed and it is possible that transcription may lead to loss of this boundary, allowing increased interactions between the 5' part of SYNE1 with the rest of the locus including ESR1. 5C contact maps and interaction counts for the control gene PUM1 (Kılıç et al, 2014) were similar in MCF7 and MDA-MB-231 cells (Fig S1D and E).

To further explore the chromatin landscape of the four ER-regulated genes, we analyzed ChIP-seq data (Guertin et al, 2014) for histone posttranslational modifications (PMT) and RNA polymerase 2 (RNA Pol2) for the two cell lines. The data show that RNA Pol2 was present at all representative genes in MCF7 but absent in MDA-MB-231 cells, except at the CCND1 gene which was constitutively active in MDA-MB-231 (Figs 1D and S1A, F, and G). The H3K27ac and H3K4me3 chromatin modifications were present at the transcription start site (TSS) and/or ERBS of ER-dependent genes in MCF7 but largely absent at genes silent in MDA-MB-231 (Figs 1D and S1A, F, and G). The H3K27me3 and H3K9me3 repressive chromatin marks largely covered the gene bodies and surrounding regulatory domains in MDA-MB-231 (Figs 1D and S1F and G). Seven ERBS exist within the TAD1—PGR domain including the 100-kb coding region and an ~300 kb upstream regulatory, intergenic region (Fig 1D). ERBS are reminiscent of enhancers and their chromatin was modified by H3K27ac in MCF7, a PMT absent at the PGR ERBSs in MDA-MB-231, except at the last, the seventh, ERBS (Fig 1D). Contacts seen in 5C appeared to occur mainly between the PGR gene body and distal upstream enhancers but not between enhancers and the TSS of the gene itself (Fig S1B). The ESR1 gene domain comprises three ER

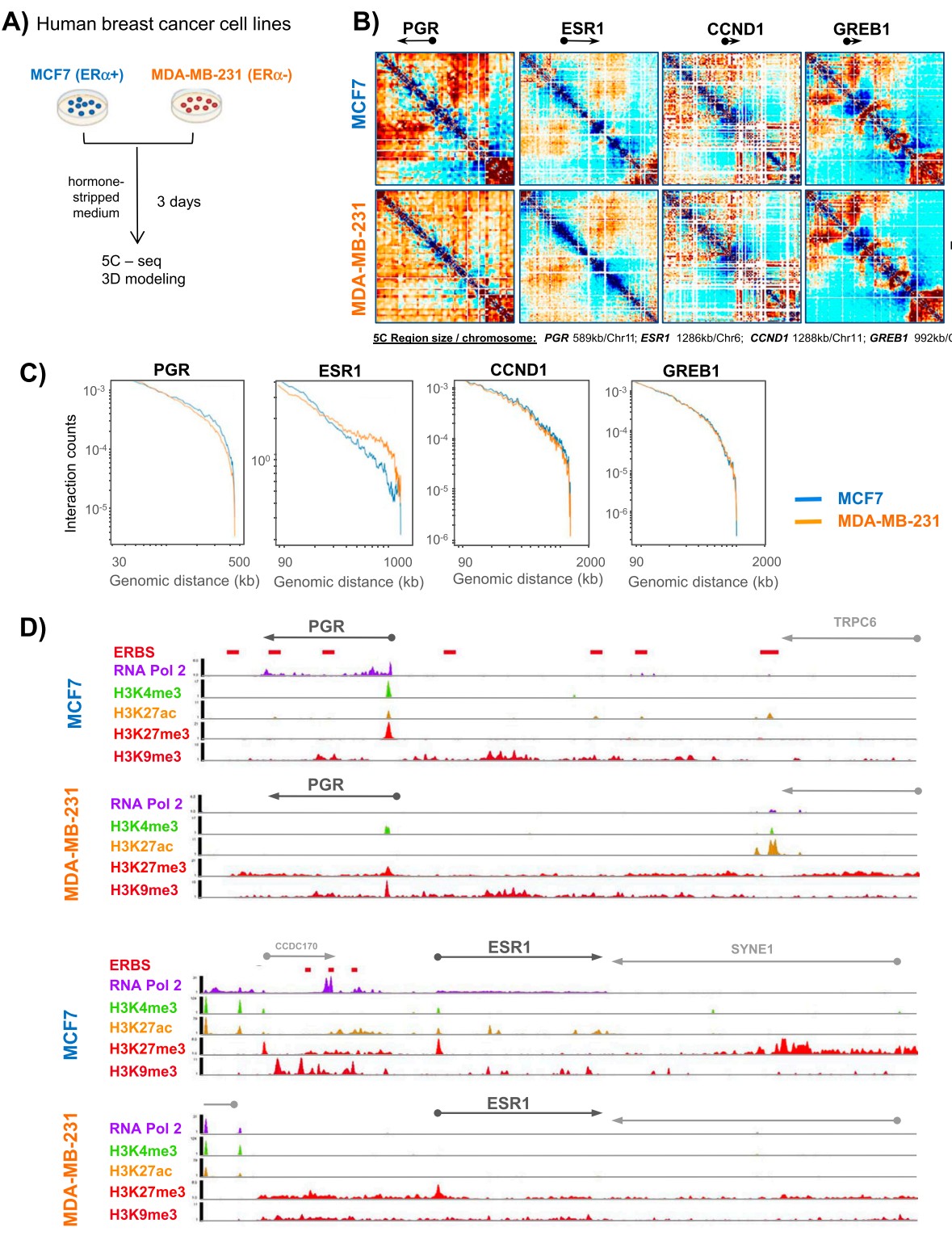

**Figure 1.   Preestablished genome organization of estrogen-regulated gene domains reflects transcriptional status.**
**(A)** Human breast cancer cell preparations and experimental flow used to interrogate 3D genome organization in cell lines expressing or not estrogen receptor α.
**(B)** Interaction frequency 5C heatmaps at 1.8 kb resolution of 0.2–1.3 Mbp domains surrounding the *PGR*, ESR1, CCND1, and GREB1 genes in MCF7 and MDA-MB-231 cells.
**(C)** Genomic interaction counts of the selected gene domains decayed by genomic distance, comparison between MCF7 (blue line) and MDA-MB-231 (red line) cells.
**(D)** Chromatin landscape of the progesterone receptor gene and the ESR1 gene domains in MCF7 and MDA-MB-231 cells, from ENCODE (ENCODE Project Consortium, 2012; Guertin et al, 2014; Luo et al, 2020).

binding sites (ERBS) upstream of the ESR1 gene. ERBS contacted each other in MCF7 cells but these interactions were not detected in the hormone-independent MDA-MB-231 cells (Fig S1C). H3K27me3 peaks were found at all promoters in both cell lines, whereas this mark only covered the gene bodies of ESR1 and *PGR* silenced in MDA-MB-231 cells (Fig 1D), consistent with broader contact frequencies (Fig 1B). The cell type-specific contact maps suggested that the 3D conformation of these gene domains relates to transcriptional competence.

### Preestablished 3D domain architecture stabilizes in response to estradiol stimulation of *PGR* transcription

To further elucidate how chromatin folding is linked to transcriptional status, we focused on the *PGR* gene. We generated 5C contact maps of the *PGR* domain from MCF7 cells grown in a hormone-stripped medium before and after 45 min and 3 h of adding 100 nM E2 (Fig 2A and B). Transcription of *PGR* increased threefold after 3 h of incubation in the presence of E2 (Kocanova et al, 2010a; Dalvai & Bystricky, 2010). We found that the overall domain architecture, in particular, the boundary next to the 3′end of the TPCR6 gene, was maintained after hormone addition (Fig 2B). Visual inspection of 5C contact maps suggested that chromatin interactions between the TSS of the *PGR* and the gene body and its downstream region (ERBS1, ERBS2 and ERBS3) were lost 3 h after E2 stimulation (Fig 2B, red arrowheads). In addition, upon early response to E2 several distinct contacts were reinforced between the TSS and proximal upstream region of the *PGR* notably between TSS–ERBS4 (Fig 2B and D). Gain of contacts between distal upstream enhancers (ERBS5, ERBS6, and ERBS7) were also observed upon 3 h of E2 stimulation (Fig 2B, green arrowhead).

To validate the specific interactions, we quantified contact frequencies between all ERBSs and between TSS-ERBSs (Fig 2C and D). We found that the interactions between ERBS3–ERBS5, ERBS3–ERBS6, and ERBS3–ERBS7 were slightly reduced at 45 min before declining at 3 h. A drop in interactions was also detected at the boundaries of the TAD1 domain, between ERBS1–ERBS7 (Fig 2C, lower panel). Two downstream (ERBS1–ERBS2 and ERBS2–ERBS3) and upstream (ERBS5–ERBS6) regions established interactions at 45 min which were reinforced at 3 h. Moreover, interactions between distal upstream enhancers of the *PGR* gene (ERBS6–ERBS7) appeared as soon as 45 min of E2 stimulation and remained stable at 3 h E2 (Fig 2C, upper panel). Notably, the promoter (TSS) of the gene did not form detectable contacts with ERBSs except with ERBS4 (Fig 2D). We noticed that TSS–ERBS4 contacts were strongly reinforced after 45 min E2 and, similar to ERBS1–ERBS2 and ERBS5–ERBS6, this interaction persisted over 3 h E2 (Fig 2C and D). Immediately after 45 min of E2 addition contacts between TSS–ERBS3, the regulatory element located within the gene body, were lost (Fig 2D). We conclude that enhancer interactions within the regulatory regions of *PGR* are modulated in an early response to hormone addition and reinforced during prolonged stimulation.

### Estrogen activation increases distal enhancer–enhancer interactions, and stabilizes folding of the *PGR* domain

To investigate the chromatin architecture of the *PGR* domain in situ, we analyzed the spatial relation between the *PGR* promoter and its

enhancers (ERBSs) by 3D DNA FISH (Kocanova et al, 2018). Fosmid probes were selected according to availability corresponding to ERBSs and the TSS of the *PGR* gene (Fig 3A). We performed 3D DNA FISH on MCF7 cells before and after 45 min and 3 h of 100 nM E2 stimulation and we measured the inter-probe distances using homemade script (plug in) running on ImageJ (Fig 3B). 3D distance measurements between pairs of fosmid probe signals were plotted (Figs 3C and S2A). In general, upon E2 activation, we noted large variations in distances measured. Distances spread from 20 to 1000 nm for fosmids separated by genomic distances from 78 to 227 kb (Figs 3C and S2A). Variations of inter-probe distances −/+ addition of E2 were not significant population-wide. When focusing on distances ≤200 nm, which represent instances detectable by 5C (Giorgetti & Heard, 2016), inter-probe measures between Fos4-Fos5 and Fos3-Fos4, located in the upstream domain, 100 and 250 kb from the *PGR*-TSS increased significantly (Figs 3D and S2B). For example, for Fos4-Fos5 and Fos3-Fos4, the proportion of measurements ranged from 4% and 19% in E2-non-treated cells to 23% and 29% in 3 h E2 treated cells, respectively (Figs 3D and S2B). In contrast, distances ≤ 200 nm significantly diminished between Fos3, covering the TSS, and Fos2, a fosmid probe located within the *PGR* gene body. We observed a decrease of short inter-probe distances between Fos2-Fos3 from 24% in untreated cells compared with 14% after 3 h E2 stimulation (Fig 3D). Distances between Fos3 (TSS) and Fos1 located in the downstream domain of the *PGR* gene were also significantly changed, with 50% of the contacts at distances ≤200 nm lost 3 h after E2 stimulation (Fig 3D). Inter-probe distances <200 nm at the PUM1 control locus did not vary between E2 untreated and treated MCF7 cells (Fig S2C and D). Variations in inter-probe distances from 3D DNA FISH observations correlate with changes in contact frequencies seen in the same areas in 5C matrices. Both by imaging and 5C, we measured significant changes of interaction between the TSS and the upstream and downstream regulatory regions of *PGR* (Figs 2B–D and 3 and S2A and B).

The conformational changes within the *PGR* domain indicated that the upstream regulatory region folds back upon itself and the *PGR* gene body upon transcription induction of *PGR* by E2. To characterize the folding behavior of multiple connected mobile genomic loci we developed an analysis, called 3-loci, based on triangulation of relative distances between three labeled sites. 3-loci is based on measurements of the distribution of 3D distances between three loci computed in single cells. Within the plane defined by the three loci, we consider that each locus moves in a circular region with radius R. So, $Sz = Sz(R)$ defining a 2D survival zone (Sz) with one degree of freedom. Using the measured distances, inverse mathematical modeling predicts each locus' distribution inside survival zones (see the Materials and Methods section). Survival zone radii of loci correlate with relative freedom of movement of one locus with respect to the two others (Lassadi et al, 2015). Here, we used three fosmids (Fig 3E) hybridizing to the *PGR* gene and two ERBS within the up and downstream regulatory regions simultaneously. In each nucleus, three distances were measured in 3D and aggregated to determine their survival zones. In cells treated with E2, survival zones of the two labeled loci (Fos1 and Fos5) surrounding the *PGR* gene were largely reduced compared with the zones these segments can explore in hormone-deprived cells. In contrast, the centrally located Fos3 locus which

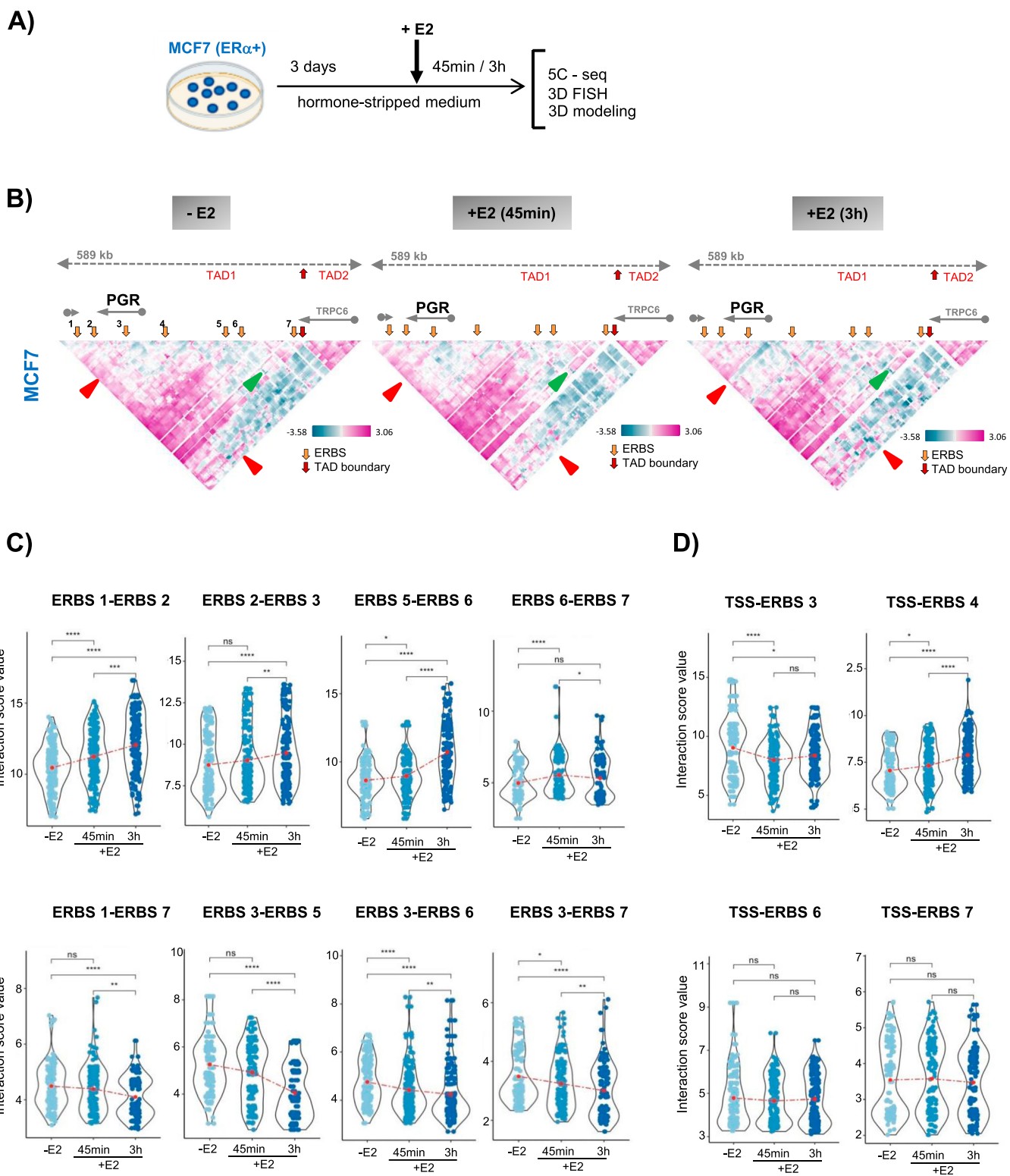

**Figure 2. Ground-state domain architecture of the progesterone receptor gene domain stabilizes, and contact frequencies between regulatory elements are modulated, in response to estradiol signaling.**
**(A)** Experimental flow used to interrogate 3D genome organization in MCF7 cells treated or not with E2. **(B)** Interaction frequency 5C heatmaps of the progesterone receptor gene domain in estrogen starved (−E2) and stimulated (+E2) for 45 min and 3 h MCF7 cells. TAD boundaries do not change in MCF7 cells (red arrows) and ESRBs are indicated (orange arrows). Increased (green arrowhead) and lost (red arrowhead) contact frequencies are indicated. **(C)** Quantification of estrogen receptor binding site (ERBS–ERBS) interactions is calculated from 5C heatmaps. Representing the gain (first row) and lost (second row) of interactions between specific ERBSs.
**(D)** Quantification of ERBS–transcription start site interactions showing variation in interaction with different ERBSs.

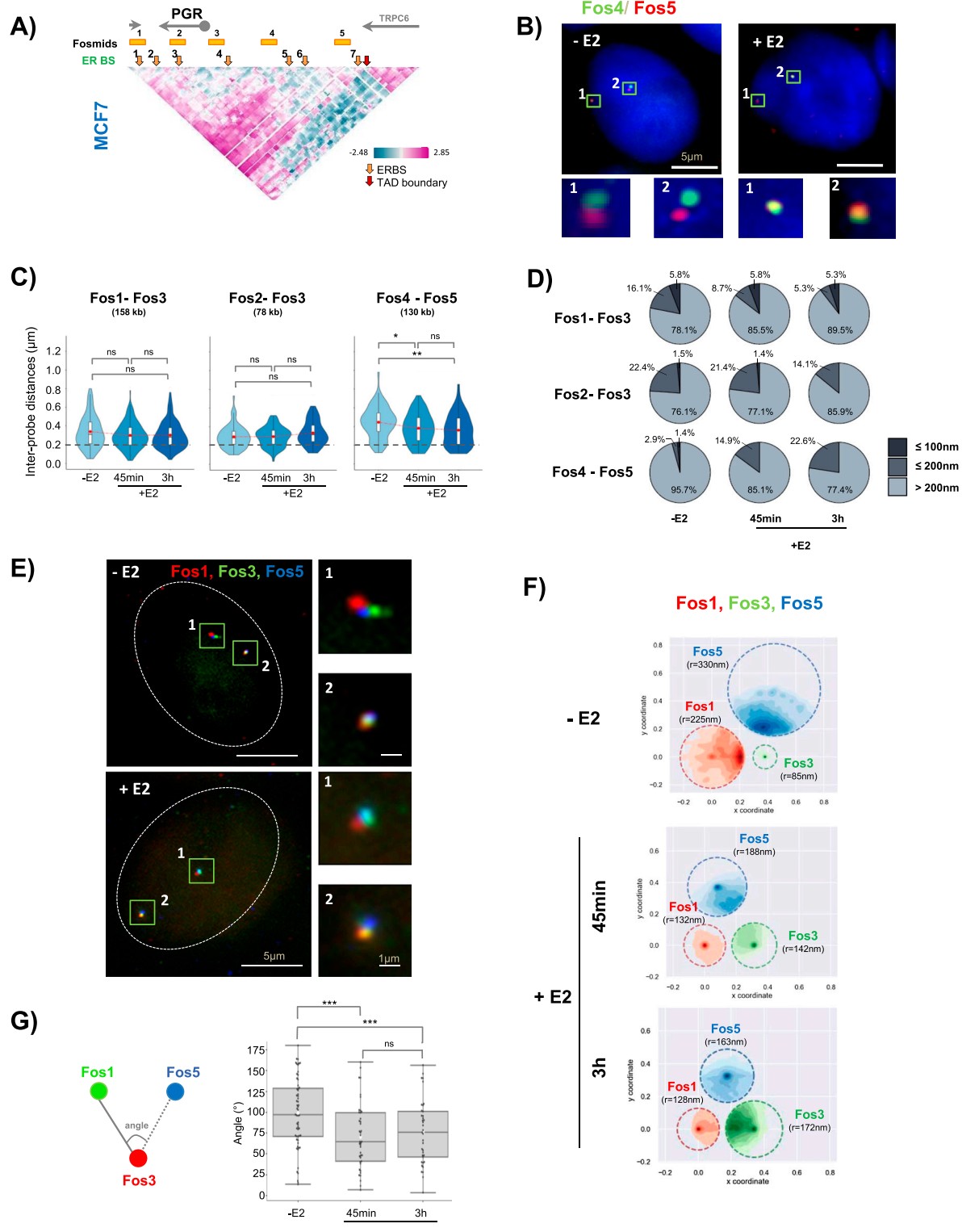

**Figure 3. Interactions between regulatory elements of the progesterone receptor gene domain are re-enforced in response to estradiol signaling in MCF7 cells.**
**(A)** Genomic position of fosmid probes used for 3D DNA FISH analysis. **(B)** Representative images from 3D imaging of dual-fosmid labeled and DAPI co-stained nuclei of MCF7 cells stimulated (+E2) or not (–E2) with 100 nM estradiol. Fos4 labeled with Alexa488 (green), Fos5 in red (labeled with Atto 647). Maximal projection of three planes is presented (0.2 μm per single plane). See MM for details. Scale bar 5 μm. **(C)** The violin plots representing inter-probe distances for different pairs of fosmids (n = 80–130 nuclei). Fisher's test: *P-values: >0.05 (ns), <0.05 (*), <0.01 (**), <0.001 (***), <0.0001 (****). **(D)** The pie charts illustrating the proportion of inter-probe distances >200 nm, interval from 200–100 nm (≤200 nm) and interval from 0–100 nm (≤100 nm) for different pairs of fosmids in MCF7 cell stimulated or not by E2. Fisher's test was used for

corresponds to the TSS region of the *PGR* seemed to become more dynamic after transcription activation by E2. Upon activation, the distal upstream region (Fos5) folds toward the *PGR* gene body. Its freedom of movement is reduced twofold (r = 330 nm to r = 163 nm, Fig 3F). The dynamics were decreased and this decrease was maintained over time (Fig 3F, 3 h), demonstrating that several enhancer elements colocate within the same cell to stabilize folding of the domain. The angle around the TSS was reduced from ~100° to <60° after E2 addition (Fig 3G) further confirming folding back of the distal upstream region over the gene body. Concomitantly, *PGR*-TSS (Fos3) freedom of movement was increased twofold, from r = 85 nm to r = 172 nm, observed 3 h poststimulation (Fig 3F).

In MCF7 cells, these results were coherent with the notion that folding is highly dynamic and variable from cell to cell (Cheng et al, 2020). Amplitude of variation in 3D positions appeared to be reduced as folding becomes stabilized when gene expression is stimulated by E2.

### Preexisting 3D structure of the *PGR* gene domain and regulatory region reorganizes during transcription activation

To further explore the structural properties of the *PGR* domain, we generated 3D models based on the spatial constraints measured by 5C contact frequencies using TADbit (Serra et al, 2017) (Fig 4A). Specifically, each region of interest was represented as a chain of spherical beads each spanning 50 nm in diameter and containing 5 kb of DNA. In MCF7 cells, representations of an overlay ~10,000 models of the 500-kb active domain of *PGR* and enhancer region revealed a "croissant"-shaped volume (Fig 4A, domains shaded purple to pink). The adjacent TPCR6 gene formed a globular structure separated physically from the *PGR* domain as expected from seeing two distinct TADs in the 5C matrices in MCF7 cells (Fig 4A, red colored domain). Restraint-based modeling enabled extrapolating distances (d) between chosen fragments (Fig 4A). Gray arches in Fig 4B are drawn between fragments which were separated by less than 50 nm in at least 50% of the calculated models of the *PGR* domain in MCF7 cells grown in the hormone-starved medium (−E2) for 3 d. Genomic sites for which distances were shortened in at least 50% of E2-treated cells (+E2) are linked by red arches (Fig 4B). This finding was coherent with a reduction in the 3D distances measured by 3D DNA FISH (Fig 3C). Distances deduced from 5C-based models and three-way 3D DNA FISH fell within the same distribution (Figs 2C and D, 3E, and 4B and S3A) although the models tended toward smaller values. Obtaining the proper scale of models is not trivial for restraint-based modeling using 3C datasets (Trussart et al, 2015). The results confirm that the preexisting *PGR* domain structure was reorganized upon estradiol treatment. In particular, numerous enhanced contacts demonstrate that RNA Pol2-bound upstream ERBS sites fold toward the gene body after 3 h E2 exposure. Enhanced contacts are reminiscent

of shortened distances measured in 3D DNA FISH and the reduced dynamics of the loci relative to each other seen by "3 loci" analysis (Fig 3E–G). Overall, the two arms of the modeled "croissant" aligned and folded upon themselves (Fig 4A). Strikingly, the adjacent TPCR6 TAD folding was not remodeled by estradiol. As expected from 5C data (Fig 1), *PGR* domain folding and hormone-induced reorganization was specific to MCF7 cells. Indeed, models from 5C data in MDA-MB-231 cells yielded distance distributions reminiscent a closed chromatin conformation of silent genes and, as expected, was not significantly altered after exposure to estradiol (Fig S3B).

### Progressive ERα binding to enhancers mediates domain folding and transcription of *PGR*

We next investigated how transcription factors and cofactors operate within this 3D landscape. In MCF7 cells, expression of the *PGR* gene can be induced within minutes by adding E2. The ensuing mRNA synthesis is known to increase over time (Shang et al, 2000; Dalvai & Bystricky, 2010; Guertin et al, 2014) suggesting multiple steps of regulation. E2 binding to the ERα triggers a conformational change of the receptor enabling it to bind to its cognate binding sequences (ERBS). Fig 5A shows progressive enrichment of ERα on all seven ERBSs within the *PGR* domain. Several chromatin-associated activators (GATA3, c-Fos, c-Jun, MYC, FoxA1), necessary for ERα-dependent gene activation were also recruited to the enhancers (Fig 5B). In contrast, ERα did not associate with the TSS of *PGR* at any of the tested time-points before and after hormone addition. However, even in hormone-starved cells, the TSS was H3K27 acetylated and bound by MYC (Fig 5B) suggesting that the TSS is primed for transcription activation in MCF7 cells. Addition of E2 rapidly led to accumulation of MYC and of RNA Pol2 at the TSS. The binding profile of RNA Pol2 at the TSS and the *PGR* gene body varied over time. We thus calculated the RNA Pol2 pausing index and quantified mRNA synthesis using RNAseq and GROseq datasets (Fig 5C). We observed a 1.5-fold (GROseq data) and a sixfold (RNAseq data) increase of RNA within the early response (45 min) of E2 stimulation compared with the control situation. The RNA Pol2 peak declined rapidly from 5 to 40 min (+E2), a time window during which *PGR* was modestly transcribed (Fig 5C, early response). From 40 to 160 min after E2 addition to the cells (late E2-response), *PGR* mRNA accumulation increased 4.5 times and RNA Pol2 was released from the TSS (Fig 5C-late response). Finally, *PGR* expression levels stabilized until 160 min post induction concomitant with reduced RNA Pol2 pausing (Shang et al, 2000; Honkela et al, 2015).

We propose that recruiting ERα to enhancers may enable enhancer interactions and conformational changes of the domain during steady state activation without direct enhancer–promoter contact. ERα accumulation thus correlates with RNA Pol2 release from a paused state and increased mRNA production (Figs 5A and C and 6B). ER-bound distal enhancer activity and chromatin looping hence fine-tune transcriptional output and permit discriminating

statistics. **(E)** Representative images from 3D imaging of triple-fosmid–labeled and DAPI co-stained nuclei of MCF7 cells stimulated (+E2) or not (−E2) with 100 nM estradiol. Maximal projection of three planes for Fos1 (in red), Fos3 (in green), and Fos5 (in blue) are presenting. Scale bar 5 μm. **(F)** Single-cell analysis of relative position of three loci simultaneously representing as survival zone distribution of Fos1, Fos3, and Fos5. See MM for details. **(G)** Angle around the Fos3 measured from mutual position of the three loci in single cell. Fisher's test was used for statistics.

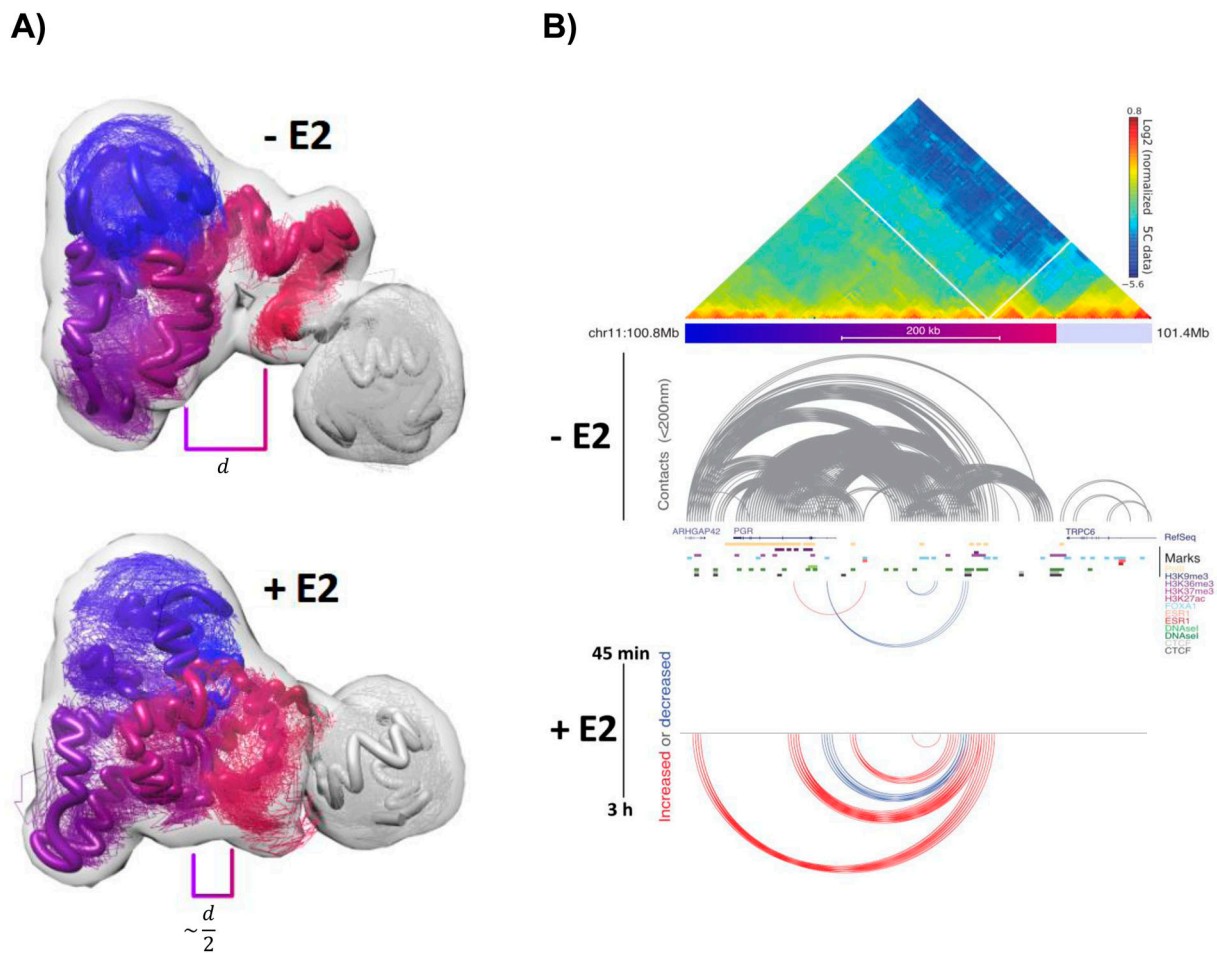

**Figure 4. 3D folding of the progesterone receptor gene (*PGR*) chromatin domain reorganizes in response to transcription activation.**
**(A)** 3D models 5C contact frequencies in MCF7 cells exposed (+E2) or not (−E2) to estradiol, 20% of the most frequent models using TADbit (Serra et al, 2017) with distances (d) between the *PGR* gene body and its upstream ERE region are displayed. **(A, B)** 5C map (normalized counts) of the *PGR* domain on Chr 11. Color bar below the map corresponds to domains in (A). **(A)** Distances derived from models shown in (A): gray arches link segments <200 nm in the maps, red and blue arches link segments for which distances increased (red) or decreased (blue) in at least 50% of the cell population after 45 min or 3 h of estradiol exposure.

transient responses from long-term sustained activity (Fig 6A and B).

## Discussion

Here, we show that chromatin fiber folding reflects transcriptional activity at the level of genomic domains and single genes. We discovered that domains encompassing silent genes show disordered structures that lack domain boundaries in triple-negative human breast cancer cell lines reminiscent of silent domains in multiple cell lines (Cheng et al, 2020). In ERα-positive MCF7 cells, the same gene domains, namely *PGR* and ESR1, display TADs containing the gene body and its enhancer-spiked regulatory region, even in the absence of transcription. This preexisting 3D structure is internally reorganized in response to hormone-induced transcription activation without altering the TAD borders. Reorganization appears to lock in already existing interactions rather than creating new ones, a phenomenon we could qualify as "caging in." In particular, at the *PGR* gene, the domain including the gene and the enhancer

region is rapidly caged in. Caging is coherent with constrained motion of an E2-induced gene (Germier et al, 2017 *Preprint*) and more generally with transcription-induced reduced chromatin dynamics (Nagashima et al, 2019; Di Stefano et al, 2020; Shaban et al, 2020) (our unpublished observations in MCF7 cells). Concomitantly to caging, ERα accumulates at numerous enhancers within the region within less than 1 h. These enhancers being close in space, ERα, and associated transcription factors and cofactors create hubs. These hubs likely correspond to ERα foci seen across the nucleus in response to E2 stimulation of hundreds of genes (Kocanova et al, 2010b). Similar to ERα, progressive binding of the glucocorticoid receptor to distal enhancers within gene domains was proposed to lead to structural reorganization (Stavreva et al, 2015). Ligand-bound nuclear receptors hence seem to induce folding between cis-acting enhancers and the adjacent gene domain. The resulting productive transcriptional conformation is similar to sustained active chromatin 3D hubs formed by the locus control region of the globin genes in erythroid cells (Kooren et al, 2007) and by Sox2 (Stadhouders et al, 2017 *Preprint*) but here as a mechanism to reversibly regulate acute activation.

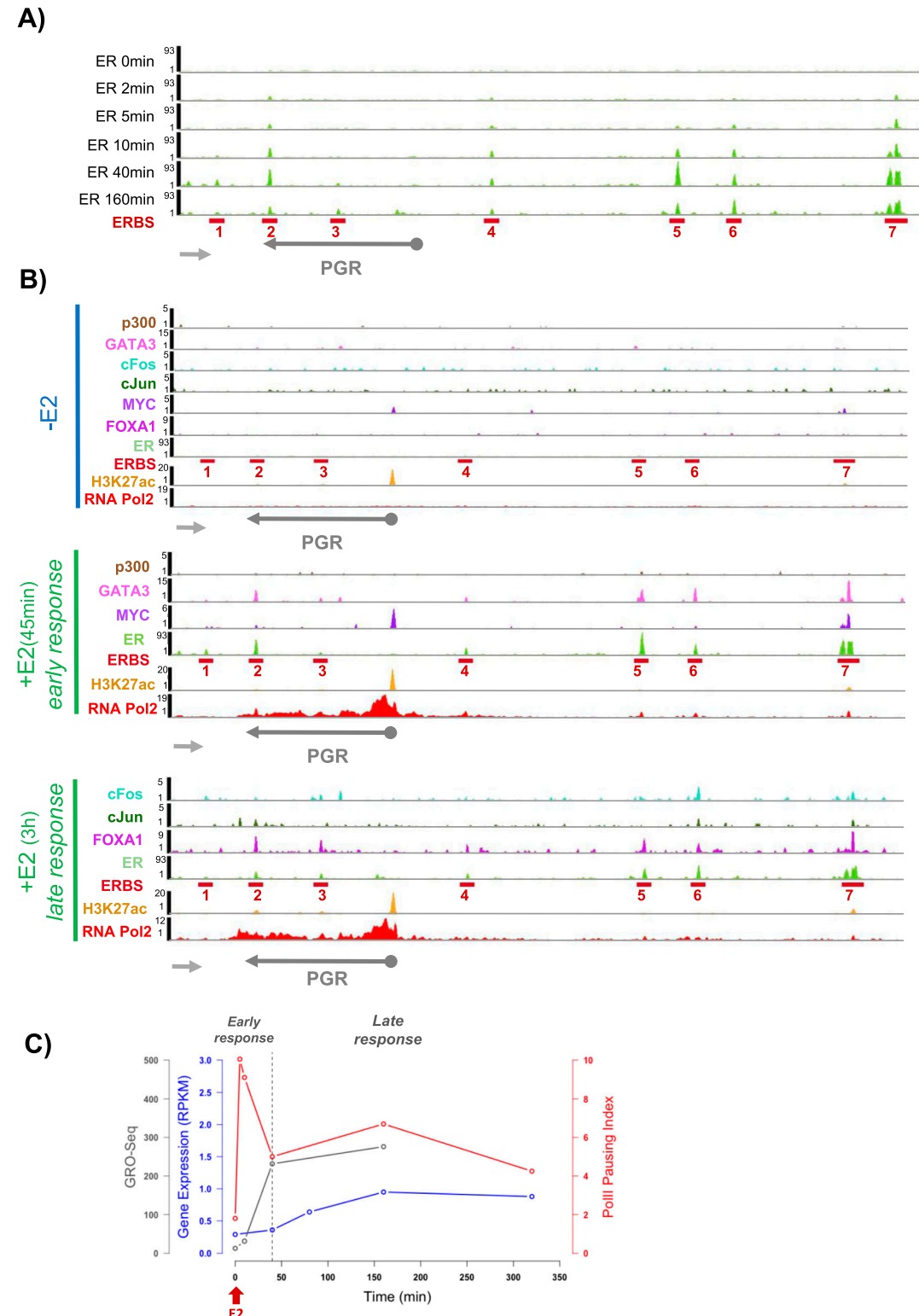

**Figure 5. Domain folding facilitates enhancer function within the progesterone receptor gene chromatin domain during estradiol induced RNA polymerase 2 releases and activation.**

**(A)** Kinetics of estrogen receptor α progressive binding on estrogen receptor binding sites within the regulatory domain of the progesterone receptor gene. Data analyzed from Guertin et al (2014). **(B)** Time-course of chromatin-associated activators (GATA3, P300, c-Fos, c-jun, MYC, FoxA1), necessary for estrogen receptor α-dependent gene activation showing their presence at early or late E2 response. **(C)** Time-course after estradiol addition to hormone-starved MCF7 cells indicating RNA Pol2 pausing index (in red), mRNA production measured by RNA-seq (in blue) and by GRO-seq (in gray).

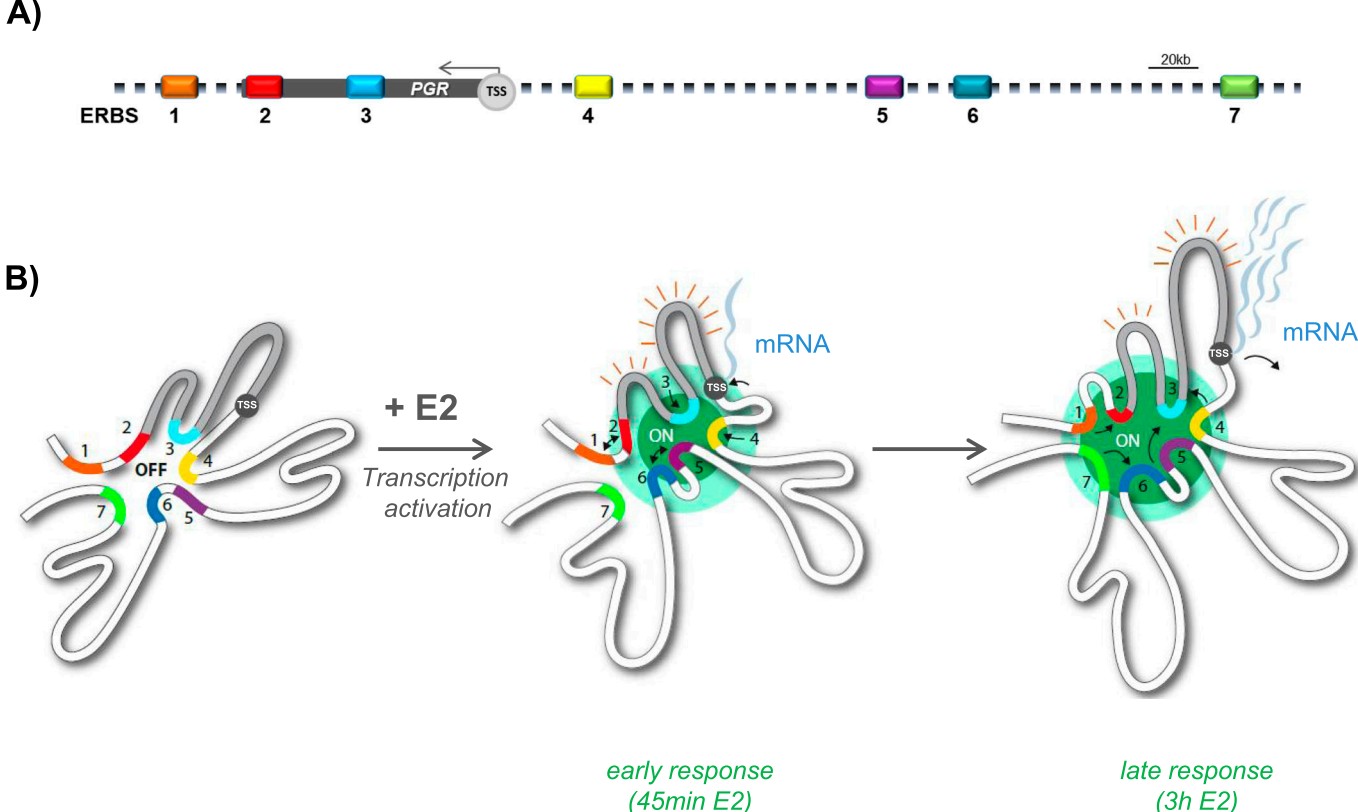

**Figure 6. 3D progesterone receptor gene (*PGR*) domain-folding steps.**
**(A)** Schematic cartoon showing linear region over 500 kb with the *PGR* gene body (100 kb), transcription start site (TSS) and seven ER binding sites (ERBS). **(B)** Model of two-state conformations of the self-interacting *PGR* domain showing the representative contacts between proximal and distal enhancers (estrogen receptor binding sites) and transcription start site upon early and late estrogen response in MCF7 cells.

Accumulation of ERα to enhancers may enable enhancer interactions and conformational changes of the domain during steady state activation (Figs 3 and 4). ERα accumulates progressively to reach a maximum concomitantly to the time of RNA Pol2 pause release and increased mRNA production (Fig 5A). Hence, ER-bound distal enhancer activity and chromatin looping fine-tune the transcriptional output.

We found that ERα exclusively binds to non-promoter sequences of the *PGR* gene domain and not to the promoter of the *PGR* gene, an observation which challenges the common view that ERα is first recruited to the promoter of target genes where it triggers recruitment of cofactors and RNA Pol2. Genome-wide binding of ERα to non-promoter sequences was reported many years ago (Carroll et al, 2006), but its role was thus far not fully appreciated. In fact, numerous other transcription factors associate with distal regulatory elements rather than promoters directly (Shlyueva et al, 2014). It appears that multistep transcription factor binding to multiple enhancers and recruitment of co-factors creates a highly sensitive and reactive system to regulate transcription.

In addition to ERα, transcription factors including GATA3 and MYC were also recruited to enhancers within the *PGR* domain. GATA3 is a zinc-finger–containing transcription factor important for cell differentiation and dedifferentiation, in particular during EMT-like transition in metastatic breast cancer (Theodorou et al, 2013).

GATA3 cooperates with ERα and is required to render cis-acting enhancers accessible for ER-mediated transcription activation (Theodorou et al, 2013; Tanaka et al, 2020). GATA3 binding was proposed to establish chromatin looping before activation (Theodorou et al, 2013) consistent with our observations that the *PGR* domain topology reflects transcriptional competence in MCF7 cells but not in MDA-MB-231 cells, which do not express GATA3. Moreover, GATA3 mutants were seen to disrupt regulatory networks enabling ER-mediated transcriptional response (Takaku et al, 2018) and repression of *PGR*.

Our study shows that 3D folding dynamics can be assessed using the 3-loci method to determine survival zones of linked genomic loci as a direct approach to analyze stimulus induced changes of specific domains without the use of sequencing-based chromosome conformation capture methodologies. We have demonstrated here that 3D models based on chromosome conformation capture data confirm the picture drawn from quantitative 3D imaging of specific loci. The 3-loci method is applicable at any scale and to any cell type.

We propose that recruiting ERα to enhancers may enable enhancer interactions and topological changes of the domain during steady state activation without direct enhancer–promoter contact. ER-bound distal enhancer activity and chromatin looping fine-tune transcriptional output and may permit discriminating transient

responses from sustained activity (Fig 6B). Preestablished chromatin architectures control gene expression without the need for de novo long-range rewiring of contacts. In fact, the common breast cancer cell lines used here may represent states of genome adaptation to optimize proliferation and response to physiological environments. An attractive hypothesis could thus be that selective estrogen receptor modulator antiestrogens exploit such an environment to hijack preset gene regulatory domains. It may therefore be relevant to explore, and possibly act upon, 3D domain organization when therapeutic resistance or recurrence appears (Fukuoka et al, 2022).

## Materials and Methods

### Cell culture

The human ER$\alpha$-positive breast cancer cell line MCF7 and the ER$\alpha$-negative breast cancer cell line MDA-MB-231 were purchased from ATTC and maintained in DMEM F-12 (Gibco) for MCF7 or DMEM (Gibco) for MDA-MB231 with Glutamax and complemented with 50 $\mu$g/ml gentamicin, 1 mM sodium pyruvate, and 10% FCS. Cells were grown at 37°C in a humidified atmosphere containing 5% $CO_2$.

To study the effects of 17$\beta$-estradiol (E2) on estrogen-regulated genes (*PGR*, ESR1, CCND1, and GREB1) cells were grown for 3 d in phenol red-free media supplemented with 10% charcoal-stripped FCS (−E2) and subsequently treated with 100 nM E2 (Sigma-Aldrich) for the indicated times.

### DNA-FISH

3D DNA FISH experiments were performed as previously described in Kocanova et al (2010a and 2018). Briefly, cells were grown for 3 d on 10-mm round glass coverslips in 24-well plates using DMEM (DMEM/F-12 for MCF7 cells) without phenol red, containing 10% charcoal-stripped FCS, before addition of 100 nM E2 for the indicated times. Coverslips were then washed once with PBS, fixed in freshly made 4% pFA/PBS for 10 min at RT and during the last 3 min a 200 $\mu$l of 0.5% Triton X-100/PBS were added homogenously. From this step, the cells were treated as followed, at RT and with moderate shaking. Cells were washed three times for 3 min in 0.01% Triton X-100/PBS, incubated in 0.5% Triton X-100/PBS for 10 min at RT and treated or not with 0.2 mg/ml RNase A in 2xSSC for 30 min at 37°C. After three washes of 10 min in PBS, cells were incubated in 0.1 M HCl for 5 min (for MCF7 cells) or in 0.05 M HCl for 2 min (for MDA-MB231cells), washed twice in 2xSSC for 3 min and then left in 50% formamide/2xSSC (pH = 7.2) 1 h minimum at RT or 1 wk maximum at 4°C before being used for 3D DNA FISH.

Fosmids were purchased at C.H.O.R.I. (see Table S1) and were labeled using the BioPrime DNA Labeling System from Invitrogen by incorporation of fluorochrome-conjugated nucleotides Atto647N-dUTP-NT (Jena Bioscience), atto550-dUTP-NT (Jean Biosciences) or ChromaTide AlexaFluor 488-5-dUTP (Molecular Probe). Labeled probes were then purified trough a G50 column from the Illustra MicroSpin G50 kit (GE Healthcare) and precipitated overnight with salmon sperm DNA (Sigma-Aldrich), human Cot-1 DNA (Invitrogen),

3 M NaAc, and 100% cold EtOH. After centrifugation, the pellet was resuspended in an appropriate volume of HP (Hybridization Premix, containing 50% formamide in a buffer solution) to obtain equal concentration (100 ng/$\mu$l) for all fosmid probes. For DNA-FISH, 200 ng of labeled fosmids were added to the prepared coverslips. The coverslips were sealed with rubber cement (Electron Microscopy Sciences) and placed into a hybridizer (Dako). Denaturation of the probes and target DNA was performed simultaneously at 85°C for 2 min and samples were then incubated overnight at 37°C. Before microscopy, coverslips were then washed, with gentle shaking, four times for 3 min in 2xSSC at 45°C, and four times for 3 min in 0.1x SSC at 60°C before being mounted with 4 $\mu$l of Vectashield Antifade Mounting Medium with DAPI at 1.5 $\mu$g/ml and sealed with transparent nail polish.

### Image acquisition and analysis

3D-DNA FISH observations were performed using an Olympus IX-81 wide-field fluorescence microscope, equipped with a CoolSNAP HQ CCD camera (Photometrics), a Polychrome V monochromator (Till Photonics) equipped with a 150 W xenon source used with a 15-nm bandwidth, an electric PIFOC piezo stepper (PI) with an accuracy of 10 nm, and imaged through an Olympus oil immersion objective 100x PLANAPO NA1.4. Acquisitions of DAPI, Alexa 488, ATTO550, and ATTO647 fluorophores were performed using multiband dichroic mirrors (Chroma), specific single-band emission filters mounted on a motorized wheel (PI), and emission filters ET450/40, ET520/40, ET580/40, and ET685/60. Image acquisitions were performed in ~21 focal planes with a 200-nm step size for 3D DNA FISH. This configuration was driven by MetaMorph (Microscopy Automation and Image Analysis Software from Molecular Devices). Images were analyzed using a home-made script running on ImageJ (Kocanova et al, 2018). Inter-probe distances were determined from 80–130 nuclei for each experimental condition and the significance of any difference in the data distributions was assessed using Fisher's test. A *P*-value ≤ 0.05 was considered statistically significant.

### ChIP-seq processing

Published ChIP-seq data in MCF7 were downloaded from GEO DataSets (https://www.ncbi.nlm.nih.gov/geo/) and treated as subsequently described. Raw data were downloaded with fastq-dump from sratoolkit (2.8.2-1). (https://github.com/ncbi/sra-tools). The quality of the reads was estimated with FastQC (0.11.7). Sequencing reads were aligned to the reference human genome assembly (hg19/GRCh37) using Burrows–Wheeler aligner (Li & Durbin, 2010; Honkela et al, 2015) (0.7.17) with default parameters. ER peak calling was performed using MACS2 (Zhang et al, 2008). ERBS were defined according to ER peak calling in +E2 40 min condition. Bigwig files were generated and normalized (RPKM) using scripts from deepTools utilities (3.0.2) (Ramírez et al, 2016). Paused RNA Pol2 indices were defined as the ratio of RNA Pol2 (total) density in the promoter–proximal region (−30 bp to TSS from +300 bp) to the total RNA Pol2 density in the transcribed regions (TSS +300 bp to TES). 5C matrices were visualized using WashU Epigenome Browser (Li et al, 2019). ER cofactors and

epigenetic marks were defined and analyzed according to published data: Table S2.

### RNA-seq processing

RNA-Seq time course in MCF7 before and after E2 treatment were downloaded from GEO (accession number: GSE62789) (Honkela et al, 2015). Raw data were downloaded with fastq-dump from sratoolkit (2.8.2-1). The quality of the reads was estimated with FastQC (0.11.7). Alignment was performed using STAR (2.6.0c) to the hg19/GRCh37 reference human genome assembly. Gene expression values were quantified from RNA sequencing data using HTSeq (0.9.1) and were then normalized (RPKM) in both cell lines.

### 5C primer design

5C primers were designed using My5C (Lajoie et al, 2009). 5C employs two types of primers: 5C forward and 5C reverse primers. We used an alternating primer where a single 5C primer was designed for each HindIII fragment throughout the genomic regions analyzed here, so that forward and reverse primers alternate (see Dostie et al (2006)).

### 5C data processing

We used My5C (Lajoie et al, 2009) to design 5C primers, using default settings. We designed 5F forward and reverse primers using the alternating primer design (Dostie et al, 2006). 5C data analysis consists of alignment, noise removal, scaling, binning, and balancing (iterative correction). First, we aligned the 5C sequencing reads to the reference primer set using Novoalign (version 3.02.00) to determine the interactions between primer pairs. Any primer pair or individual primer that has very low or excessive number of interactions will be removed from the matrix using z-score in both cis and trans using thresholds 6 and 12, respectively. Then the matrix was read-normalized to adjust for the number of sequencing reads per sample. Finally, we binned and balanced the matrix to decrease bias, complexity, and get multi-resolution data using the ICE method (Imakaev et al, 2012): Data were binned in 15-kb bins with eight steps which creates 13.125 kb overlaps between bins. Balancing uses Sinkhorn–Knopp algorithm which rescales the rows and columns by dividing their sums by their means iteratively to get matrix convergence. Thus, the total number of interactions per primer (locus) should be the same. LOWESS method (Locally Weighted Regression: An Approach to Regression Analysis by Local Fitting) was used to estimate the expected interactions for given distances.

### 5C data analysis

Interaction counts by genomic distance for estrogen-regulated gene regions (*PGR*, ESR1, CCND1, and GREB1) have been computed with hicPlotDistVsCounts tool from HiCExplorer (2.2.1.1) utilities. Correlation Heatmaps were computed with hicCorrelate tool from HiCExplorer. To study interaction counts distributions at ERBS/ERBS and ERBS/TSS for the *PGR* gene, we considered a window region from each ER peak summit ±10 kb (±15 kb for ERE7)

within the *PGR* gene region matrices and *PGR* gene TSS ±10 kb. We then plotted the corresponding averaged interaction count values around the two interacting regions, and compared each of the two E2 induction time-point distributions. Violin plots have been generated using R home-made scripts.

### Modeling

5C produces 2D matrices that represent the frequency of interactions between loci along the genomic region of interest. To transform such data into a 3D conformation of higher-order chromatin folding, we used TADbit (Serra et al, 2017). Structure determination by TADbit can be seen as an iterative series of three main steps: translating the data into spatial restraints, constructing an ensemble of structures that satisfy these restraints, and analyzing the ensemble to produce the final structure. Specifically, each particle pair in the models was restrained by a series of harmonic oscillator centered on a distance derived from the 5C data as previously described (Dostie et al, 2006; Serra et al, 2017). A total of 1,000 models at 5 kb resolution per region were built by TADbit with *maxdist = 250, upfreq = 0, lowfreq = −0.6, scle = 0.01* as input parameters. The Pearson correlation coefficient between a contact map obtained from the optimal models and the input 5C matrix was 0.94, indicative of accurate models (Trussart et al, 2015). All remining TADbit *model* parameters were set to default values. The resulting models were further analyzed to obtain contact arches analysis, distance distributions, and visual representations of the models.

#### *Contact arches*
A "contact arch" was defined between two model particles if their distance was less than 200 nm in at least 50% of the models in the ensemble. To assess the number of increased/decreased contacts, we computed the difference in models having such arch. An increase of contacts in at least 20% of models with respect to −E2 models indicated an increase of an arch (red color). Conversely, a decrease of contacts in at least 20% of models with respect −E2 models indicated a decrease of an arch (blue color).

#### *Distance analysis*
Euclidean distances in Cartesian space between selected fosmids were calculated for the entire ensemble and represented as box plots. If a fosmid occupied more than one particle, the center of mass of the constituting particles was used as the *x,y,z* coordinates of the fosmid.

#### *Ensemble visualization*
The UCSF Chimera package (Yang et al, 2012), a highly extensible program for interactive visualization of molecular structures, was used to produce all images of the models. Models were superimposed and visualized with a wire representation within a transparent molecular density map of the occupancy of all particles in the models (obtained using the *molmap* function in Chimera). Finally, the "centroid" model (that is, the model central to the superimposed ensemble) was represented as a thick worm-

like structure for easy visualization (obtained using the *shape tube* function in Chimera).

### Computing survival radii—3 loci analysis

Distances between the three loci R, G, and B were measured in 3D (see above).

$$((R - G)_i)_{i=1..n}, ((G - B)_i)_{i=1..n} et ((R - B)_i)_{i=1..n} \tag{1}$$

In the plane defined by the three loci, we denote by $d_{rgmin}$, $d_{rbmin}$, $d_{gbmin}$, the minimal distances) and $d_{rgmax}$, $d_{rbmax}$, $d_{gbmax}$ the maximal distances between R–G–B, respectively.

We can note that

$$d_{rgmin} \leqslant min(((R - G)_i)_{i=1..n}) \tag{2}$$

$$d_{rgmax} \geqslant max(((R - G)_i)_{i=1..n}) \tag{3}$$

We have the same inequalities for couples ($d_{rbmin}$, $d_{rbmax}$) and ($d_{gbmin}$, $d_{gbmax}$).

In the limit case, we make the assumption that when **n** is large enough,

$$d_{rgmin} = min(((R - G)_i)_{i=1..n}) \tag{4}$$

$$d_{rgmax} = max(((R - G)_i)_{i=1..n}) \tag{5}$$

$$d_{rbmin} = min(((R - B)_i)_{i=1..n}) \tag{6}$$

$$d_{rbmax} = max(((R - B)_i)_{i=1..n}) \tag{7}$$

$$d_{gbmin} = min(((G - B)_i)_{i=1..n}) \tag{8}$$

$$d_{gbmax} = max(((G - B)_i)_{i=1..n}) \tag{9}$$

We define the following relations as follows:

$$d_{rgmax} = 2(R_r + R_g) + d_{rgmin} \tag{10}$$

$$d_{rbmax} = 2(R_r + R_b) + d_{rbmin} \tag{11}$$

$$d_{gbmax} = 2(R_g + R_b) + d_{gbmin} \tag{12}$$

Solving this equation gives the following:

$$R_g = \frac{d_{rgmax} + d_{gbmax} - d_{rbmax} - d_{rgmin} - d_{gbmin} + d_{rbmin}}{4} \tag{13}$$

$$R_b = \frac{-d_{rgmax} + d_{gbmax} + d_{rbmax} + d_{rgmin} - d_{gbmin} - d_{rbmin}}{4} \tag{14}$$

$$R_r = \frac{d_{rgmax} - d_{gbmax} + d_{rbmax} - d_{rgmin} + d_{gbmin} - d_{rbmin}}{4} \tag{15}$$

When applying the following change of variable,

$$A = \frac{d_{rbmax} + d_{rbmin}}{2} \tag{16}$$

$$B = \frac{d_{gbmax} + d_{gbmin}}{2} \tag{17}$$

$$C = \frac{d_{rgmax} + d_{rgmin}}{2} \tag{18}$$

We can specify the position of the third locus in relation to the other loci by

$$cos(\theta) = \frac{C^2 + B^2 - A^2}{2CB} \tag{19}$$

where $\theta$ is the angle between Red–Green and Green–Blue axes.

For each triplet of measured distances, $((R - G)_i, (G - B)_i, (R - B)_i)_{i=1..n}$ the positions of the loci $((x_r,y_r), (x_g,y_g), (x_b,y_b))$ were deduced as follows:

$$\begin{cases} (x_r - x_g)^2 + (y_r - y_g)^2 = (R - G)_i^2 \quad (x_g - x_b)^2 + (y_g - y_b)^2 \\ \quad = (G - B)_i^2 \quad (x_r - x_b)^2 + (y_r - y_b)^2 \\ \quad = (R - B)_i^{2'} Underconstraints' (x_r - x_O^r)^2 + (y_r - y_O^r)^2 \\ \quad \leq R_r^2 \quad (x_g - x_O^g)^2 + (y_g - y_O^g)^2 \leq R_g^2 \quad (x_b - x_O^b)^2 + (y_b - y_O^b)^2 \leq R_b^2 \end{cases} \tag{20}$$

To solve system Equation (20), the three survival zones using polar coordinates were calculated:

$$x_{ij}^c = X_0^c + R_i cos(\theta_j) \quad y_{ij}^c = Y_0^c + R_i sin(\theta_j) \text{ with } R_i = \frac{R_r}{N} i \quad \theta_j = \frac{2\pi}{M} j \tag{21}$$

where c is the color index **R**, **G** or **B**. N is the number of survival zones subdivision and M is the polar angle subdivision.

## Data Availability

Code and data can be accessed at https://github.com/FlavienRaynal/PGR_Kocanova for 5C and ChIP-seq analyses and at https://src.koda.cnrs.fr/alain.kamgoue.3/tripleloci for triple loci.

## Supplementary Information

## Acknowledgements

We acknowledge support from the Light Imaging Toulouse CBI facility (LITC). We thank David Villa (scienceimage.fr) for assistance with figure design. This work was supported by grants from HFSPO RGP0044 to J Dekker, MA Marti-Renom, and K Bystricky; Agence Nationale de la Recherche (ANR): SINFONIE AAPG-18-CE12-006 to K Bystricky. MA Marti-Renom acknowledges support from the Spanish Ministerio de Ciencia e Innovación (PID2020-115696RB-I00).

J Dekker acknowledges support from the National Human Genome Research Institute (HG003143). J Dekker is an investigator at the Howard Hughes Medical Institute.

## Author Contributions

S Kocanova: data curation, formal analysis, validation, investigation, methodology, and writing—original draft, review, and editing.
F Raynal: data curation, formal analysis, methodology, and writing—review and editing.
I Goiffon: investigation and methodology.
BA Oksuz: formal analysis and investigation.
D Baú: investigation.
A Kamgoué: software and methodology.
S Cantaloube: formal analysis.
Y Zhan: investigation.
B Lajoie: investigation.
MA Marti-Renom: conceptualization, data curation, software, formal analysis, supervision, funding acquisition, validation, investigation, methodology, and writing—original draft, review, and editing.
J Dekker: conceptualization, data curation, formal analysis, supervision, funding acquisition, validation, investigation, methodology, project administration, and writing—original draft, review, and editing.
K Bystricky: conceptualization, formal analysis, supervision, funding acquisition, validation, investigation, visualization, methodology, project administration, and writing—original draft, review, and editing.

## Conflict of Interest Statement

MA Marti-Renom receives consulting honoraria from Acuity Spatial Genomics, Inc. J Dekker is a consultant for Arima Genomics (San Diego, CA), and Omega Therapeutics (Cambridge, MA). The other authors declare no competing interests.

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
