## [Reviewer comments · Life Science Alliance]

Life Science Alliance

Enhancer-driven 3D chromatin domain folding modulates transcription in human mammary tumor cells

Silvia Kocanova, Flavien Raynal, Isabelle Goiffon, Betul Oksuz, Davide Bau, Alain Kamgoué, Sylvain Cantaloube, Ye Zhan, Bryan Lajoie, Marc A. Marti-Renom, Job Dekker, and Kerstin Bystricky

DOI: <https://doi.org/10.26508/lsa.202302154>

Corresponding author(s): Kerstin Bystricky, French National Centre for Scientific Research and Marc A. Marti-Renom,

Review Timeline:

Submission Date:	2023-05-12
Editorial Decision:	2023-07-06
Revision Received:	2023-09-29
Editorial Decision:	2023-10-23
Revision Received:	2023-11-03
Accepted:	2023-11-06

Transaction Report:

July 6, 2023

Re: Life Science Alliance manuscript #LSA-2023-02154-T

Kerstin Bystricky
CBI University of Toulouse

Dear Dr. Bystricky,

Thank you for submitting your manuscript entitled "Enhancer-driven local 3D chromatin domain folding modulates transcription in human mammary tumor cells" to Life Science Alliance. The manuscript was assessed by expert reviewers, whose comments are appended to this letter. We invite you to submit a revised manuscript addressing the Reviewer comments.

Thank you for this interesting contribution to Life Science Alliance. We are looking forward to receiving your revised manuscript.

Sincerely,

B. MANUSCRIPT ORGANIZATION AND FORMATTING:

Reviewer #1 (Comments to the Authors (Required)):

Kocanova et al present an analysis of the rewiring of 3D chromosomal architecture in response to estrogen receptor (ER) signalling in a breast cancer cell line, using an ER-negative cell line as a control and focusing in particular on the PGR locus. They present 5C and FISH data, as well as a chromatin fiber modelling based on 5C.

Overall, this an interesting and extensive analysis that merits publication in this journal. However, we found the paper quite difficult to read and the plots difficult to decipher (potentially due to the loss of resolution upon conversion by the journal's publishing system?).

General comments:

1. The authors present multiple rewiring events upon ER stimulation throughout the main text. However, they conclude that pre-existing chromosomal architecture undergoes relatively little rewiring in these conditions. It would help the reader considerably if the authors framed the narrative in a way that makes it clear that they the observed rewiring events are minor, in the authors' opinion (and ideally also why so). Also, if the authors believe that these changes are indeed minor, it may be worth reducing the amount of minute detail presented in the main text (and potentially also in the main figures), signposting more clearly the changes that are, in their opinion, particularly notable and important.
2. Currently the FISH and 5C analysis are presented alongside each other, and it is not immediately apparent how these data support each other and if they are truly in alignment (or highlight different parts of the same picture). It would help if the authors juxtaposed/integrated these results more extensively to help the reader.
3. It is critical to check that image resolution is improved upon final submission, as in the current form 5C and genome browser track plots are barely legible (quite possibly not the authors' fault!).

Specific comments:

- Figure 1:

- a) Panel A: there's a slight plotting issue with the diagram
- b) Could the authors clarify why some gene labels above the plots are drawn with a dot and arrow (PGR1, ESR1) and some with just an arrow (CCND1), and yet some with an arrow in Panel B and without one in Panel D?
- c) Panel D: the genome track plots are very small. It may be that less is more and having fewer, bigger tracks will be more helpful to the reader, with the rest moved to supplementary.

- Bottom of p5: "Within the CCND1 and GREB1 gene domains, only a few architectural features were cell type specific in agreement with relatively similar expression levels". Unclear what the authors mean, since GREB1 seems to be expressed almost exclusively in MCF7 cells (Fig 1D and S1A).

- p6: "In contrast, significant conformational differences were detected for the PGR and ESR1 gene domains. PGR and ESR1 were silent in MDA-MB-231 cells and poised for transcription in MCF7 cells (Fig. 1B)". Did the authors mean Fig 1D? Also, what is the evidence for poising in MCF7? Both K27me3 and K27ac peaks are visible (in addition to K4me3), suggesting heterogeneity rather than bivalency.

- p6-7: "ERBS are reminiscent of enhancers and their chromatin was modified by H3K27ac in MCF7, a PMT absent at the PGR ERBSs in MDA-MB-231, except at the last, the 7th, ERBS (Fig. 1D)". To validate the enhancer nature of ERBSs (as the authors proceed to refer to them hereinafter) it would be good to confirm that they are marked with H3K4me1. Is there published or in-house H3K4me1 data available for these cells? Alternatively, could the authors check in resources such as Ensemble Regulatory Build whether these regions are known to have enhancer annotations in other cell lines/types?

- p7: "Transcription of PGR increased 3 fold under these conditions (Dalvai and Bystricky, 2010; Kocanova et al., 2010a)." Could the authors specify if they refer to the 45 min and/or the 3h treatment?

- Figure 2B + text p7: There seems to be a mix-up in the meaning of the red and green arrowhead between the figure legend and the main text. The legend of figure 2B says: "Increased (red arrowhead) and lost (green arrowhead) contact frequencies are

indicated." The text says: "Visual inspection of 5C contact maps suggested that identified chromatin interactions occurred between the TSS of the PGR and the gene body as well as its down-stream region (ERBS1, ERBS2 and ERBS3) were lost 3h after E2 stimulation (Fig. 2B, red arrowheads). In addition, upon early response to E2 a few distinct contacts were reinforced between the TSS and proximal up-stream region of the PGR notably between TSS- ERBS4 (Fig. 2B and 2D). Gain of contacts between distal up-stream enhancers (ERBS5, ERBS6 and ERBS7) were also observed upon 3 h of E2 stimulation (Fig. 2B, green arrowhead)."

- Figure 3: no legend for Panel G is given and Panel H is referred to in the text (p.10) but absent on the figure.
- We found Figure 4A difficult to understand. Some pointers on the figure and/or in the legend would be helpful. Also, the authors could consider using more contrasting colours than pink and purple.
- The authors may consider discussing their use of Pol II pausing index vs Pol II-Ser5 ChIP to study Pol II pausing.

Reviewer collaboration

We have read the comments by the other two reviewers with much interest. In our view, the extension of scope suggested by Reviewer 2, would require too much time and resource to be justified for this paper.

Reviewer #2 (Comments to the Authors (Required)):

In this paper the authors use a combination of approaches to explore the changes in the three-dimensional architecture of primarily the progesterone receptor gene (PGR) in response to estrogen stimulation in the estrogen receptor positive (ER+) breast cancer cell line MCF7. They provide 5C, in situ hybridization and ChIP-seq data to support a model of PGR activation by ER that involves reinforcement of an existing 3D chromatin confirmation. The data are interesting and the studies are well done. Several changes could increase the overall impact of the work.

- 1) The authors focus almost exclusively on PGR for detailed analysis but do show some data for the ER target genes CCND1 and GREB1. It would increase the generalizability of the findings to show a more complete analysis of the changes induced by estrogen at CCND1 and GREB1.
- 2) Except for the use of MDAMB231 cells as a negative control, all of the studies of estrogen response are in MCF7 cells. It would be interesting to also show results for T47D cells which express higher levels of PGR.
- 3) The authors measure gene expression, but do not directly measure transcription. Run-on experiments using GRO-seq or a similar approach would allow the authors to more directly link the dynamics of the changes in chromatin with transcription rates.

There are a few minor points that also should be address.

- 1) On page 5 the authors describe MDAMB231 and MCF7 as "representative of key breast cancer (BC) etiologies..." I think they mean to write "breast cancer subtypes" as there is no evidence that the etiologies are different.
- 2) On page 14 the authors state that "We found that ER α almost exclusively binds to non-promoter sequences of the PGR gene domain, an observation which challenges the common view that ER α is first recruited to the promoter of target genes where it triggers recruitment of cofactors and RNApol2." As the authors point out this was established by Carroll and others many years ago. They should remove this claim.

Reviewer #3 (Comments to the Authors (Required)):

The manuscript by Kocanova, et al. summarizes extensive mapping of long-range enhancer-promoter, chromosomal interactions that underlie folding and genomewide organization of estrogen-responsive genes in cultured breast cancer cells. By comparing two cell lines, one ER+ (MCF7) and one TNBC-like (MDA-MB231) by 5C modeling in an estrogen-depleted state and ChIP-seq of histone modifications and RNA polymerase II (RNA Pol II), the authors determined that further focus on the progesterone gene (PGR) expression was most illustrative of estrogen-induced response over time of gene expression in MCF7 cells. Deeper study by quantitative 3D FISH, additional ChIP-seq of transcription factors and histone PTM at specific time points led to development of quantitative comparisons of FISH and a model of genome organization whereby pre-existing chromosomal architecture is altered by accumulation of ER and transcription factor binding over time that enables regulatory hub formation and efficient gene transcription.

The work presented in this manuscript is likely to be of some interest to the community and is based, in general, on strong supportive data. Minor revisions are needed to better support the claims made. Changes induced in chromosomal architecture and enhancer-promoter interactions have been documented by many studies over the years in the case of estrogen stimulation and, especially, using MCF7 cells. However, the current work increases our understanding of the process by deeper analyses of the PGR gene locus, especially quantitative chromatin architecture analyses combined with an additional approach of quantitative FISH over time of estrogen induction of transcription.

Specific critique:

1. The authors based their initial conclusions on comparisons of two, distinct cell lines of human tumor origin: MCF7 and MDA-MB231. Since these were originally derived from different patients and have been cultured for decades, the claim that ER-status is a major determinant of different transcription outcomes and chromosomal architecture cannot be made. The authors must make this distinction, when stating the conclusions derived from Figure 1 and add to the Discussion that results must be interpreted with this in mind. Fortunately, the studies of Figure 1 were used primarily to establish a rationale for deeper focus on the PGR gene of MCF7 cells +/- estrogen and do not detract from the generalized conclusions and specific quantitative analyses.
2. The authors state, regarding Fig. 1B, that gene domains of PGR and ESR1 genes were poised for transcription in MCF7 cells. That conclusion is not well supported until the data of later figures, especially Figure 5, are presented. The authors should wait to state this, since Pol II binding and active histone PTMS are not obvious, until the later figure is presented with better supportive data.
3. Data presented in figures following Figure 1 are well supported and of considerable interest. The 3loci approach is likely to be of interest to the community.
4. The modeling presented in Figure 4A is not particularly convincing and does not add significantly to the overall conclusions. The figure 4B, 5C map seems much more supportive and easier to interpret.
5. Figure 5 data are of considerable interest and important for the overall conclusions of the work. The progressive enrichment of ER in response to estrogen stimulation and the timing of various transcription factor binding and histone PTMs are important for overall interpretation of chromosomal architecture and impact. For these reasons, the authors must include all of the transcription factors assessed at all time points. It is not clear why only a subset is presented at different time points.
6. In the discussion, the authors state that the common view held is that ER binds to the promoter of actively transcribing, responsive genes. This is overstated, as ER has commonly been held to bind at Estrogen Regulatory Elements or EBS, which are not necessarily at the promoter. This should be restated.
7. The claim in the discussion that antiestrogens have specific outcomes begs the question of why the authors did not actually use antiestrogens to back their claims of estrogen receptor-mediated alterations in chromatin organization. This discussion should be tempered.

Dear Dr Sawey,

we thank the reviewers for evaluating our manuscript in detail and for providing valuable feedback to our work. We gladly acknowledge their enthusiastic comments on our extensive analysis of long-range enhancer-promoter, chromosomal interactions that underlie folding and genome wide organization of estrogen-responsive genes in cultured breast cancer cells. We are grateful that reviewers concur that this manuscript is likely to generate considerable interest within the scientific community due to its robust supporting data from complementary methodologies and innovative contributions, notably the introduction of novel approaches such as the 3loci analysis.

The suggestions of the reviewers helped improve our manuscript. We clarified several sections and provided additional information.

Below, please find detailed replies to all comments. Reviewers' questions are in **black**, our replies to them in **green**. We highlight added/modified sections in the revised manuscript in **yellow**.

We appreciate your assistance and eagerly await your response, along with the prospect of our work being published in Life Science Alliance.

Sincerely,

Kerstin Bystricky and co-authors

Detailed responses to the reviewers:

Reviewer #1

Kocanova et al present an analysis of the rewiring of 3D chromosomal architecture in response to estrogen receptor (ER) signalling in a breast cancer cell line, using an ER-negative cell line as a control and focusing in particular on the PGR locus. They present 5C and FISH data, as well as a chromatin fiber modelling based on 5C.

Overall, this an interesting and extensive analysis that merits publication in this journal. However, we found the paper quite difficult to read and the plots difficult to decipher (potentially due to the loss of resolution upon conversion by the journal's publishing system?).

We thank the reviewer for their enthusiastic evaluation. Indeed, conversion of some figure panels was poor. We apologize for the inconvenience and thank the reviewers for their understanding. We corrected this and hope that the panels will now be of good quality. As you will see below, we clarified a set of sections and rewrote others for more fluidity.

General comments:

1. The authors present multiple rewiring events upon ER stimulation throughout the main text. However, they conclude that pre-existing chromosomal architecture undergoes relatively little rewiring in these conditions. It would help the reader considerably if the authors framed the narrative in a way that makes it clear that they the observed rewiring events are minor, in the authors' opinion (and ideally also why so). Also, if the authors believe that these changes are indeed minor, it may be worth reducing the amount of minute detail presented in the main text (and potentially also in the main figures), signposting more clearly the changes that are, in their opinion, particularly notable and important.

We thank the reviewer for this suggestion. Several very detailed descriptions were deleted on pages 7, 8 and 9, so that we could better highlight the relevant changes in chromatin architecture to be considered by the readers.

2. Currently the FISH and 5C analysis are presented alongside each other, and it is not immediately apparent how these data support each other and if they are truly in alignment (or highlight different parts of the same picture). It would help if the authors juxtaposed/integrated these results more extensively to help the reader.

Indeed, 5C and FISH data complement each other. The used fosmids for 3D FISH and the primers for 5C do not cover exactly the same regions yet the results with both approaches are generally consistent and support the same picture. On p 11, we highlight three sentences integrating and comparing the results from orthogonal methods. A statement was included in the discussion p13.

3. It is critical to check that image resolution is improved upon final submission, as in the current form 5C and genome browser track plots are barely legible (quite possibly not the authors' fault!).

Indeed, the conversion to pdf was of very low resolution, we apologize for the inconvenience. High resolution images for figures are now provided.

Specific comments:

- Figure 1:

a) Panel A: there's a slight plotting issue with the diagram

We do not see any plotting issue in the version we submitted for publication. We will double check that the resubmission files are all legible.

b) Could the authors clarify why some gene labels above the plots are drawn with a dot and arrow (PGR1, ESR1) and some with just an arrow (CCND1), and yet some with an arrow in Panel B and without one in Panel D?

We thank the reviewer for pointing this out. This has been corrected.

c) Panel D: the genome track plots are very small. It may be that less is more and having fewer, bigger tracks will be more helpful to the reader, with the rest moved to supplementary.

A new figure 1 including minor changes is now presented. The data for CCND1 and GREB1 genes (Fig 1D) were moved to Fig S1F, S1G to make the data in the main figure better readable.

- Bottom of p5: "Within the CCND1 and GREB1 gene domains, only a few architectural features were cell type specific in agreement with relatively similar expression levels". Unclear what the authors mean, since GREB1 seems to be expressed almost exclusively in MCF7 cells (Fig 1D and S1A).

This was corrected. In both cell lines the genes are expressed. They are activated in MCF7, and constitutively transcribed (to different levels, but not silent as PGR) in MDA-MB231.

- p6: "In contrast, significant conformational differences were detected for the PGR and ESR1 gene domains. PGR and ESR1 were silent in MDA-MB-231 cells and poised for transcription in MCF7 cells (Fig. 1B)". Did the authors mean Fig 1D? Also, what is the evidence for poising in MCF7? Both K27me3 and K27ac peaks are visible (in addition to K4me3), suggesting heterogeneity rather than bivalency.

The finding is that in MCF7 cells the PGR and ESR1 domains are organized even in the absence of hormone prior to transcription activation (Fig.1B), an organization that is not seen when the genes are silent in MDA-MB231 cells. In agreement, when comparing chromatin modifications (Fig 1D), H3K27 acetylation is already present in the absence of hormone at these genes in MCF7 cells and absent in MDA-MB231 cells. The description was reformulated in the manuscript to reflect this.

- p6-7: "ERBS are reminiscent of enhancers and their chromatin was modified by H3K27ac in MCF7, a PMT absent at the PGR ERBSs in MDA-MB-231, except at the last, the 7th, ERBS (Fig. 1D)". To validate the enhancer nature of ERBSs (as the authors proceed to refer to them hereinafter) it would be good to confirm that they are marked with H3K4me1. Is there published or in-house H3K4me1 data available for these cells? Alternatively, could the authors check in resources such as Ensemble Regulatory Build whether these regions are known to have enhancer annotations in other cell lines/types?

We have checked the presence of H3K4me1 at the PGR region in MCF7 cells (Chip-seq from ENCODE public database, Figure R1). We present the analysis for your examination here. The tracks for H3K4me1 and H3K27ac for the region surrounding the PGR demonstrate that both epigenetic marks overlap with defined EREs. Thus, we trust that these can be considered as enhancer regions.

Figure R1. Chromatin landscape showing the presence of H3K4me1 and H3K27ac around the PGR gene domain in MCF7 cells (Chip-seq from ENCODE public database).

- p7: "Transcription of PGR increased 3 fold under these conditions (Dalvai and Bystricky, 2010; Kocanova et al., 2010a)." Could the authors specify if they refer to the 45 min and/or the 3h treatment?

We refer to 3 hours of E2 treatment. This information was added in the manuscript (p7).

- Figure 2B + text p7: There seems to be a mix-up in the meaning of the red and green arrowhead between the figure legend and the main text. The legend of figure 2B says: "Increased (red arrowhead) and lost (green arrowhead) contact frequencies are indicated." The text says: "Visual inspection of 5C contact maps suggested that identified chromatin interactions occurred between the TSS of the PGR and the gene body as well as its down-stream region (ERBS1, ERBS2 and ERBS3) were lost 3h after E2 stimulation (Fig. 2B, red arrowheads). In addition, upon early response to E2 a few distinct contacts were reinforced between the TSS and proximal up-stream region of the PGR notably between TSS-

ERBS4 (Fig. 2B and 2D). Gain of contacts between distal up-stream enhancers (ERBS5, ERBS6 and ERBS7) were also observed upon 3 h of E2 stimulation (Fig. 2B, green arrowhead)."

We thank the reviewer for pointing out this mistake, which has now been corrected.

- Figure 3: no legend for Panel G is given and Panel H is referred to in the text (p.10) but absent on the figure.

We thank the reviewer for pointing this out. Indeed, the panel H does not exist for the figure 3. Indication of 3h corresponds to 3 hours treatment. This was changed in the text.

The legend for panel G has been added.

- We found Figure 4A difficult to understand. Some pointers on the figure and/or in the legend would be helpful. Also, the authors could consider using more contrasting colours than pink and purple.

In the revised manuscript, we are providing an improved version of the Fig 4.

Color bar from blue to red is the common one used for structural biology. The contrast is not best from purple and pink just because the distance we are contrasting is between two regions close in sequence. If we use a different color map, the problem would still persist, for example yellow to red, the contrast would be between orange and dark orange. We found the presented range the best given these limitations.

- The authors may consider discussing their use of Pol II pausing index vs Pol II-Ser5 ChIP to study Pol II pausing.

We acknowledge this interesting suggestion. Unfortunately, Pol2Ser5 ChIP data are not available for the MCF7 cell line before and after E2 treatment. It has been shown that computing the Pol2 Pausing Index using global Pol2 is robust and reliable (see the initial paper (Adelman & Lis, 2013) and many other papers thereafter). Thus, we consider that in our case using the Pol2 Pausing index calculation based on global Pol 2 is robust.

Reviewer collaboration

We have read the comments by the other two reviewers with much interest. In our view, the extension of scope suggested by Reviewer 2, would require too much time and resource to be justified for this paper.

We thank this reviewer for her/his discernment.

Reviewer #2

In this paper the authors use a combination of approaches to explore the changes in the three-dimensional architecture of primarily the progesterone receptor gene (PGR) in response to estrogen stimulation in the estrogen receptor positive (ER+) breast cancer cell line MCF7. They provide 5C, in situ hybridization and ChIP-seq data to support a model of PGR activation by ER that involves reinforcement of an existing 3D chromatin confirmation. The data are interesting and the studies are well done. Several changes could increase the overall impact of the work.

1) The authors focus almost exclusively on PGR for detailed analysis but do show some data for the ER target genes CCND1 and GREB1. It would increase the generalizability of the findings to show a more complete analysis of the changes induced by estrogen at CCND1 and GREB1.

The four gene domains are different. Our study did not aim to generalize but to determine a molecular and structural mechanism of transcriptional status and regulation. A detailed analysis of the CCND1 and GREB1 gene would be complex given current methodologies: the CCND1 gene is only 14 kb long, and the GREB1 gene is surrounded by other genes whose contribution to 5C maps may not be neutral. CCND1 and GREB1 are expressed in both cell lines and the data shown do not reveal significant differences in the organization of the surrounding domain. Our previously published FISH data show enhancer-promoter contact modulation for the CCND1 gene (Kocanova et al. Methods 2018). The ESR1 gene comprises several promoters and its transcriptional regulation differs from the one of PGR: after a first activation it is subsequently repressed. Here we compared PGR and ESR1 domains in MDA-MB231 cells (silent) and MCF7 cells (active) to illustrate that folding of domains around silent genes are relatively unremarkable and do not change under the conditions we studied.

2) Except for the use of MDAMB231 cells as a negative control, all of the studies of estrogen response are in MCF7 cells. It would be interesting to also show results for T47D cells which express higher levels of PGR.

MDA-MB231 and MCF7 are distinct cell lines derived from different patients and tumours. Our study focuses on the transcriptional status of PGR and not on the properties of the cell lines. Moreover, the T47D cell line expresses higher levels of PGR because it harbors 4 copies of the gene rather than 2. For example, G-banding analysis clearly shows only 2 copies of chr11 in MCF7 and 4 copies in T47D (Fig 3 in <https://doi.org/10.1016/j.bbrc.2017.03.114>). Therefore, the allele variability between MCF7 and T47D would be an additional layer of complication for both 5C and FISH data analysis presented here, which would be difficult to interpret given the additional variability these multiple copies could generate.

3) The authors measure gene expression, but do not directly measure transcription. Run-on experiments using GRO-seq or a similar approach would allow the authors to more directly link the dynamics of the changes in chromatin with transcription rates.

We thank the reviewer for this relevant suggestion. We analyzed the GRO-Seq dataset from Hah et al., 2011 measuring Neo-Nascent RNA genome-wide in MCF7 cells at four time points - 0, 10, 40 and 160 min after E2 stimulation. This dataset allowed us to analyze PGR transcription rates in early and beginning of late responses. The new GRO-Seq analysis is now included in Fig 5C, along with RNA-seq quantification and the RNA Pol2 pausing index.

There are a few minor points that also should be address. 1) On page 5 the authors describe MDAMB231 and MCF7 as "representative of key breast cancer (BC) etiologies..." I think they mean to write "breast cancer subtypes" as there is no evidence that the etiologies are different.

This was changed.

2) On page 14 the authors state that "We found that ER α almost exclusively binds to non-promoter sequences of the PGR gene domain, an observation which challenges the common view that ER α is first recruited to the promoter of target genes where it triggers recruitment of cofactors and RNApol2." As the authors point out this was established by Carroll and others many years ago. They should remove this claim.

We agree that ER is indeed known to bind to EREs/EBS, which are not necessarily at the promoter. In fact, ER- ChIA-Pet analysis shows that 90% of ER binding and contacts occur in intergenic non promoter regions (see below Figure R2 for your information). In our study we show that at the PGR gene ER does not need to bind to the promoter for activation to occur. This finding was somewhat unexpected and we felt it was different from the common view. We have now rephrased this statement to make our point clearer (p14):

Corrections in the manuscript (page 14):

We found that ER α exclusively binds to non-promoter sequences of the PGR gene domain but not to the promoter of the PGR gene, an observation which challenges the common view that ER α is first recruited to the promoter of target genes where it triggers recruitment of cofactors and RNA Pol2. Genome-wide binding of ER α to non-promoter sequences was reported many years ago (Carroll et al., 2006) but the role of this association was thus far not fully appreciated. In fact, numerous other transcription factors associate with distal regulatory elements rather than promoters directly (Shlyueva et al., 2014).

Figure R2. MCF7 ChIA-Pet analysis for gene and promoter anchoring. About half of ERE are intragenic and half are intergenic (left). However, ERE overlap with gene promoters (right) only in about 10% of the interactions. These results are in agreement with our ER ChIP-Seq analysis.

Reviewer #3

The manuscript by Kocanova, et al. summarizes extensive mapping of long-range enhancer-promoter, chromosomal interactions that underlie folding and genomewide organization of estrogen-responsive genes in cultured breast cancer cells. By comparing two cell lines, one ER+ (MCF7) and one TNBC-like (MDA-MB231) by 5C modeling in an estrogen-depleted state and ChIP-seq of histone modifications and RNA polymerase II (RNA Pol II), the authors determined that further focus on the progesterone gene (PGR) expression was most illustrative of estrogen-induced response over time of gene expression in MCF7 cells. Deeper study by quantitative 3D FISH, additional ChIP-seq of transcription factors and histone PTM at specific time points led to development of quantitative comparisons of FISH and a model of genome organization whereby pre-existing chromosomal architecture is altered by accumulation of ER and transcription factor binding over time that enables regulatory hub formation and efficient gene transcription.

The work presented in this manuscript is likely to be of some interest to the community and is based, in general, on strong supportive data. Minor revisions are needed to better support the claims made. Changes induced in chromosomal architecture and enhancer-promoter interactions have been documented by many studies over the years in the case of estrogen stimulation and, especially, using MCF7 cells. However, the current work increases our understanding of the process by deeper analyses of the PGR gene locus, especially quantitative chromatin architecture analyses combined with an additional approach of quantitative FISH over time of estrogen induction of transcription.

Specific critique:

1. The authors based their initial conclusions on comparisons of two, distinct cell lines of human tumor origin: MCF7 and MDA-MB231. Since these were originally derived from different patients and have been cultured for decades, the claim that ER-status is a major determinant of different transcription outcomes and chromosomal architecture cannot be made. The authors must make this distinction, when stating the conclusions derived from Figure 1 and add to the Discussion that results must be interpreted with this in mind. Fortunately, the studies of Figure 1 were used primarily to establish a rationale for deeper focus on the PGR gene of MCF7 cells +/- estrogen and do not detract from the generalized conclusions and specific quantitative analyses.

The statement that ERalpha is a major determinant of breast tumor growth is true in general as Breast Cancer first diagnosis still relies on ER status. It is of course also true that cell lines are distinct as described on p5 and we agree that results must be interpreted with this in mind.

2. The authors state, regarding Fig. 1B, that gene domains of PGR and ESR1 genes were poised for transcription in MCF7 cells. That conclusion is not well supported until the data of later figures, especially Figure 5, are presented. The authors should wait to state this, since Pol II binding and active histone PTMS are not obvious, until the later figure is presented with better supportive data.

We thank the reviewer for pointing this out to us. We reformulated the section at the beginning to describe that the domains are organized. The notion that this organization may indicate a poised state to facilitate transcription activation upon estradiol treatment is indeed more relevant and experimentally supported in subsequent parts of the manuscript.

3. Data presented in figures following Figure 1 are well supported and of considerable interest. The 3 loci approach is likely to be of interest to the community.

Thank you to this evaluation.

4. The modeling presented in Figure 4A is not particularly convincing and does not add significantly to the overall conclusions. The figure 4B, 5C map seems much more supportive and easier to interpret.

In the revised manuscript, we are providing an improved version of the Figure 4 with information supporting our findings by 5C and 3D DNA-FISH. The models presented in this figure show that extrapolating distances (d) between chosen fragments of the PGR domain are shorter after E2 treatment. These findings are compatible with reduced 3D distances measured by 3D DNA FISH.

5. Figure 5 data are of considerable interest and important for the overall conclusions of the work. The progressive enrichment of ER in response to estrogen stimulation and the timing of various transcription factor binding and histone PTMs are important for overall interpretation of chromosomal architecture and impact. For these reasons, the authors must include all of the transcription factors assessed at all time points. It is not clear why only a subset is presented at different time points.

Most of the analyses shown in Figure 5 are from published ENCODE data sets. ChIP-seq data for some transcription factors are only available at certain time-points. We believe that the large number of data presented, some for all time points, others only for 2 of the 3, is pertinent for the interpretation of the overall study. Adding additional tracks or datasets would result in a very complex figure, which may make its interpretation more cumbersome.

6. In the discussion, the authors state that the common view held is that ER binds to the promoter of actively transcribing, responsive genes. This is overstated, as ER has commonly been held to bind at Estrogen Regulatory Elements or EBS, which are not necessarily at the promoter. This should be restated.

We agree that ER is indeed known to bind to EREs/EBS, which are not necessarily at the promoter. In fact, ER- ChIA-Pet analysis shows that 90% of ER binding and contacts occur in non-promoter regions. Here we show that at the PGR gene ER does not need to bind to the promoter for activation to occur. The latter finding was somewhat unexpected and we felt it was different from the common view. We have now clarified this statement (p14).

Correction in the manuscript (page 14):

We found that ER α exclusively binds to non-promoter sequences of the PGR gene domain but not to the promoter of the PGR gene, an observation which challenges the common view that ER α is first recruited to the promoter of target genes where it triggers recruitment of cofactors and RNA Pol2. Genome-wide binding of ER α to non-promoter sequences was reported many years ago (Carroll et al., 2006) but the role of this association was thus far not fully appreciated. In fact, numerous other transcription factors associate with distal regulatory elements rather than promoters directly (Shlyueva et al., 2014)

7. The claim in the discussion that antiestrogens have specific outcomes begs the question of why the authors did not actually use antiestrogens to back their claims of estrogen receptor-mediated alterations in chromatin organization. This discussion should be tempered.

We thank the reviewer for this suggestion. We tempered the discussion.

Correction in the manuscript (page 15):

Preestablished chromatin architectures control gene expression without the need for de-novo long range rewiring of contacts. In fact, the common breast cancer cell lines used here may represent states of genome adaptation to optimize proliferation and response to physiological environments. It may

therefore be relevant to explore, and possibly act upon, 3D domain organization when therapeutic resistance or recurrence appears (Fukuoka et al., 2022)

October 23, 2023

RE: Life Science Alliance Manuscript #LSA-2023-02154-TR

Prof. Kerstin Bystricky
French National Centre for Scientific Research
CBI
118 route de Narbonne
Université Paul Sabatier
Toulouse 31062
France

Dear Dr. Bystricky,

Thank you for submitting your revised manuscript entitled "Enhancer-driven 3D chromatin domain folding modulates transcription in human mammary tumor cells". We would be happy to publish your paper in Life Science Alliance pending final revisions necessary to meet our formatting guidelines.

- please upload your primary and supplementary figures as single files
- please add ORCID ID for the secondary corresponding author--they should have received instructions on how to do so
- please note that titles in the system and on the manuscript file must match
- please use the [10 author names et al.] format in your references (i.e., limit the author names to the first 10)
- please add your main, supplementary figure, and table legends to the main manuscript text after the references section
- please incorporate supplementary references into the main references section in the manuscript text
- please upload your Tables in editable .doc or excel format
- please add a callout for Figure 6A-B to your main manuscript text

A. FINAL FILES:

B. MANUSCRIPT ORGANIZATION AND FORMATTING:

Sincerely,

Reviewer #1 (Comments to the Authors (Required)):

I thank the authors for addressing my comments and recommend that the manuscript should be accepted.

Reviewer #3 (Comments to the Authors (Required)):

In this resubmission, Koncanova et al. present a much clear model of estrogen-regulated, long-range enhancer interactions, chromosomal architecture and transcription control. Each of my previous concerns were adequately addressed.

November 6, 2023

RE: Life Science Alliance Manuscript #LSA-2023-02154-TRR

Prof. Kerstin Bystricky
French National Centre for Scientific Research
CBI
118 route de Narbonne
Université Paul Sabatier
Toulouse 31062
France

Dear Dr. Bystricky,

Thank you for submitting your Research Article entitled "Enhancer-driven 3D chromatin domain folding modulates transcription in human mammary tumor cells". It is a pleasure to let you know that your manuscript is now accepted for publication in Life Science Alliance. Congratulations on this interesting work.

DISTRIBUTION OF MATERIALS:

Again, congratulations on a very nice paper. I hope you found the review process to be constructive and are pleased with how the manuscript was handled editorially. We look forward to future exciting submissions from your lab.

Sincerely,
